# Equivalent Linear Mappings of Large Language Models

**James R. Golden**                                                    *jamesgolden1@gmail.com*
*Oakland, CA*

**Reviewed on OpenReview:** *https://openreview.net/forum?id=oDWbJsIuEp*

## Abstract

Despite significant progress in transformer interpretability, an understanding of the computational mechanisms of large language models (LLMs) remains a fundamental challenge. Many approaches interpret a network's hidden representations but remain agnostic about how those representations are generated. We address this by mapping LLM inference for a given input sequence to an equivalent and interpretable linear system which reconstructs the predicted output embedding with relative error below $10^{-13}$ at double floating-point precision, requiring no additional model training. We exploit a property of transformer decoders wherein every operation (gated activations, attention, and normalization) can be expressed as $A(x) \cdot x$, where $A(x)$ represents an input-dependent linear transform and $x$ preserves the linear pathway. To expose this linear structure, we strategically detach components of the gradient computation with respect to an input sequence, freezing the $A(x)$ terms at their values computed during inference, such that the Jacobian yields an equivalent linear mapping. This "detached" Jacobian of the model reconstructs the output with one linear operator per input token, which is shown for Qwen 3, Gemma 3 and Llama 3, up to Qwen 3 14B. These linear representations demonstrate that LLMs operate in extremely low-dimensional subspaces where the singular vectors can be decoded to interpretable semantic concepts. The computation for each intermediate output also has a linear equivalent, and we examine how the linear representations of individual layers and their attention and multilayer perceptron modules build predictions, and use these as steering operators to insert semantic concepts into unrelated text. Despite their expressive power and global nonlinearity, modern LLMs can be interpreted through equivalent linear representations that reveal low-dimensional semantic structures in the next-token prediction process. Code is available at https://github.com/jamesgolden1/equivalent-linear-LLMs/.

## 1 Introduction

The transformer decoder is the architecture of choice for large language models (Vaswani et al., 2017) and efforts toward a conceptual understanding of its mechanisms are ongoing (Sharkey et al., 2025). Significant insights include sparse autoencoders for conceptual activations in LLMs (Bricken et al., 2023; Templeton et al., 2024; Lieberum et al., 2024), linear probes (Alain & Bengio, 2016), "white-box" alternative architectures (Yu et al., 2023) and analytic results on generalization (Cowsik et al., 2024). While transformers are complex globally nonlinear functions of their input, we demonstrate how to compute an equivalent linear system that reconstructs the predicted output embedding for a given input sequence up to double floating-point precision.

Our approach directly extends the framework of Elhage et al. (2021), who analyzed attention-only transformers as interpretable linear circuits, but were limited to small models without MLPs (due to gated activation functions) or normalization layers. We show that by detaching nonlinear terms from the gradient computation, modern LLMs with gated activations (as well as softmax attention and normalization) can be decomposed into an equivalent linear system for a given input. Recently, Kadkhodaie et al. (2023) showed that powerful image denoising diffusion models with ReLU activations and certain architectural constraints are piecewise linear functions which can be computed via the Jacobian and can be clearly interpreted as

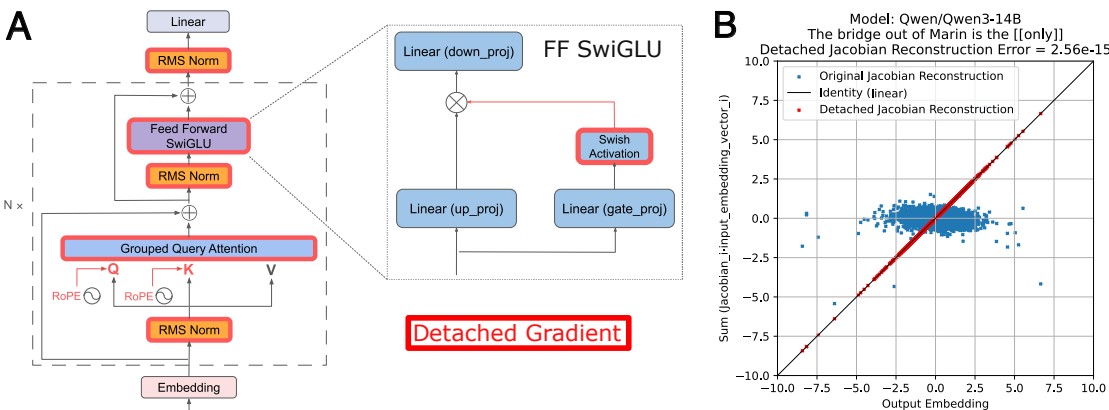

Figure 1: A) A schematic of the transformer decoder (Grattafiori et al., 2024; Nvidia, 2024). The PyTorch gradient detach operations for components outlined in red effectively freeze the nonlinear activations for a given input sequence, creating a linear path for the gradient with respect to the input embedding vectors, but do not change the output. The output embedding prediction can be mapped to an equivalent linear system by the Jacobian autograd operation. The feedforward module with a gated linear activation function is shown in expanded form to demonstrate how the gating term can be detached from the gradient to form a linear path, achieving linearity for a given input. The RMSNorm layers and softmax attention blocks also must be detached from the gradient. B) For the input sequence "The bridge out of Marin is the", the elements of the predicted output embedding vector of the model compared to the elements from the Jacobian reconstruction for both the original Jacobian (blue points) and detached Jacobian operations (red points), shown for Qwen 3 14B. Note that the detached Jacobian reconstructions match the predicted embedding, with relative error (the norm of the reconstruction error divided by the norm of the output embedding) less than $10^{-13}$ for double floating-point precision. See reconstructions for Llama 3.2 3B and Gemma 3 4B in Fig. A2.

low-dimensional adaptive linear filters with comprehensible singular vectors. For many open-weight LLMs, every component operation (gated activations, attention, and normalization) can be expressed in the form $A(x) \cdot x$, where $A(x)$ represents an input-dependent linear transform and $x$ preserves the linear pathway. The gradient operation with respect to the input can be manipulated at inference by freezing the $A(x)$ terms at their values during inference operation with the detach operation such that the output embedding prediction has a linear equivalent as in Fig. 1. This "detached" Jacobian $\mathbf{J}^+$ computation captures the complete forward operation of the model, including activation functions and attention modules, although it must be recomputed for each input sequence (as it is not piecewise linear but "pointwise" linear).

This approach allows us to analyze a model from input embeddings to predicted output embedding as an equivalent linear system for a particular input sequence. By examining the singular value decomposition (SVD) of the equivalent linear system, we can measure the local dimensionality of the learned manifolds involved in next-token prediction and can decode the singular vectors into output tokens. This analysis can also be done layer by layer, or for individual attention and multilayer perceptron (MLP) modules, in order to observe how these models compose next-token predictions.

We demonstrate equivalent linearity in model families including Qwen 3, Gemma 3, Llama 3, at a range of sizes up to Qwen 3 14B. (See the appendix for additional equivalent linear demonstrations for Deepseek R1 0528 Qwen 3 8B Distill, Phi 4, Mistral Ministral and OLMo 2). This approach offers a path to interpreting LLMs for specific inputs that could serve as a complement to other powerful interpretability methods. While this is a local method that is somewhat computationally intensive, this approach does not require additional training as required for sparse autoencoders. For example, training sparse autoencoders for Gemma 2 9B (Lieberum et al., 2024) required substantial compute across multiple feature widths and layers, and must be repeated for each new model and layer. Our approach works immediately on LLMs with gated activations and zero-bias linear layers, and produces a more exact representation for interpretation than other methods.

If equivalent linear mapping were applied to next-token prediction at scale, this would offer a form of interpretability as the difficult but tractable problem of analyzing many equivalent linear systems.

## 2 Method

### 2.1 The Jacobian of a deep ReLU Network

Mohan et al. (2019) observed that deep *ReLU* networks for image denoising which utilize zero-bias linear layers are "adaptive linear" functions due to their homogeneity of order 1 at a given fixed input, which enables interpretation as an equivalent linear system. Given the homogeneity at a fixed input, the network's output can be reproduced by numerically computing the Jacobian matrix of the network at a particular input image $\mathbf{x_{im}^*}$ and multiplying it by $\mathbf{x_{im}^*}$.

$$\mathbf{y_{im}^*} = \mathbf{J}(\mathbf{x_{im}^*}) \cdot \mathbf{x_{im}^*} \tag{1}$$

Due to the global nonlinearity of the network, the Jacobian must usually be computed again at every input of interest. The Jacobian may be the same for similar inputs in the same piecewise region of the response (Balestriero & Baraniuk, 2021; Black et al., 2022) (but this will be demonstrated to not be the case for transformer architectures).

### 2.2 The Jacobian of a transformer decoder

Many open weight LLMs also use linear layers with zero bias, as required for linearity in the architecture of Mohan et al. (2019). A transformer decoder predicts an output token embedding $\mathbf{y}$ given a sequence of $k$ input tokens $\mathbf{t} = (\mathbf{t_0}, \mathbf{t_1}..., \mathbf{t_k})$ mapped to input embedding vectors $\mathbf{x} = (\mathbf{x_0}, \mathbf{x_1}..., \mathbf{x_k})$, where $\mathbf{t}^*$ and $\mathbf{x}^*$ represent a particular sequence. The output embedding prediction is a nonlinear function of the input embedding vectors $\mathbf{x_0}, \mathbf{x_1},...\mathbf{x_k}$, as LLMs utilize nonlinear gated activation functions for layer outputs (SwiGLU for Llama 3, GELU for Gemma 3 and Swish for Qwen 3) as well as normalization and softmax attention blocks.

Gated activations like $Swish(\mathbf{x}) = \mathbf{x} \cdot sigmoid(\beta \cdot \mathbf{x})$, with a linear term and a nonlinear term, are also an "adaptive" linear function or, more generally, an adaptive homogeneous function of order 1 (Mohan et al., 2019). If the $sigmoid(\beta \cdot \mathbf{x})$ term that gives rise to the nonlinearity is frozen for a specific numerical input, e.g. an embedding vector $\mathbf{x_0^*}$ (Elhage et al., 2021) (or equivalently detached from the computational graph with respect to the input), then we have a linear function valid only at $\mathbf{x_0^*}$ where (1) holds and we can numerically compute a Jacobian matrix that carries out $Swish(\mathbf{x_0^*})$ as a linear operation.

Below we show that computing the Jacobian after effectively substituting specific values for the nonlinear terms also works for other gated activation functions, normalization layers and softmax attention blocks. We further demonstrate that for a given input sequence we can apply necessary gradient detachments so that the entire transformer decoder is an adaptive homogeneous function of order 1, and numerically compute the equivalent linear system that reproduces the transformer output embedding $\mathbf{y}^*$.

The Jacobian $\mathbf{J}(\mathbf{x})$ of a transformer is the set of matrices generated by taking the partial derivative of the decoder inference function $\mathbf{y}(\mathbf{x}) = f(\mathbf{x_0}, \mathbf{x_1}..., \mathbf{x_k})$, with respect to each element of each $\mathbf{x_i}$ (where $\mathbf{x_i}$ for Llama 3.2 3B has length 3072, for example, and therefore the Jacobian matrix for each embedding vector is a square matrix of this size). If a transformer decoder were naturally a homogeneous function of order 1, this Jacobian would generate an equivalent representation of the network.

However, this is not the case. In order to numerically compute an equivalent linear representation, we introduce a "detached" Jacobian $\mathbf{J}^+$, which is a set of matrices that captures the full nonlinear forward computation for a particular input sequence $\mathbf{x}^*$ as a linear system. The detached Jacobian is the numerical Jacobian of the LLM forward operation when its gradient includes a specific set of $detach()$ operations for the nonlinear terms in the normalization, activation and attention operations that force the function to be "adaptively" homogeneous of order 1. The detached Jacobian operates on its corresponding input embedding vector to provide a reconstruction of the LLM forward operation (shown in Fig. A1 and validated in Fig. 1B

by the PyTorch "allclose" function for absolute and relative tolerances of $10^{-13}$).

$$\mathbf{y}^* = \sum_{i=0}^{k} \mathbf{J_i^+}(\mathbf{x}^*) \cdot \mathbf{x_i^*} \tag{2}$$

The conventional Jacobian $\mathbf{J}$ for a particular input sequence $\mathbf{x}^*$ (as in Mohan et al. (2019)) does not generate an accurate reconstruction the nonlinear LLM forward operation since the transformer function is not homogeneous of order 1. The detached Jacobian $\mathbf{J}^+$ evaluated at $\mathbf{x}^*$ is the result of an alternative gradient path through the same network which is homogeneous with respect to the input $\mathbf{x}^*$. The detached Jacobian $\mathbf{J}^+$ only generates an accurate reconstruction at $\mathbf{x}^*$ and not in the local neighborhood due to the strong nonlinearity of the decoder inference function. The detached Jacobian matrices differ for every input sequence and must be computed numerically for every sequence.

### 2.3 Nonlinear layers as linear operators for a given input

In order to achieve linearity, modifications must be made to the gradient computations of the RMSNorm operation, the activation function (SwiGLU in Llama 3.2) and the softmax term in the attention block output.

#### 2.3.1 Normalization

Normalization layers like LayerNorm (Xu et al., 2019) or RMSNorm (Zhang & Sennrich, 2019) with zero bias are nonlinear with respect to their input because they include division by the square root of the variance of the input.

$$norm(\mathbf{x}) = \frac{\mathbf{x}}{\sqrt{var(\mathbf{x})}} \tag{3}$$

Mohan et al. (2019) devised a novel bias-free batch-norm layer which detaches the variance term from the network's computational graph (see their code implementation). Their batch-norm layer returns the same values as the standard batch-norm layer, but it is linear at inference as the nonlinear operation is removed from the gradient computation. This is also similar to the "freezing" of nonlinear terms in attention-only transformers from Elhage et al. (2021).

We make a similar change for Llama 3.2 3B by altering how the gradient with respect to the input is computed at inference for RMSNorm. This is accomplished by substituting the value for the input vector $\mathbf{x}^*$ for only the variance term as in (4). In PyTorch, this is accomplished by cloning and detaching the $\mathbf{x}$ tensor within the variance operation, so its value will be treated as a constant. The gradient operation is still tracked for $\mathbf{x}$ in the numerator, so that term will be treated as a variable by the PyTorch autograd function for computing the Jacobian. The gradient of the function is then computed at $\mathbf{x}^*$ (we assume for simplicity an input sequence of length 1).

$$norm(\mathbf{x}) = \frac{\mathbf{x}}{\sqrt{var(\mathbf{x}^*)}} \tag{4}$$

We define the detached Jacobian as follows:

$$\mathbf{J_n^+} = [\frac{\partial}{\partial \mathbf{x}} norm(\mathbf{x})]|_{\mathbf{x}=\mathbf{x}^*} \tag{5}$$

We can rewrite the pointwise linear RMSNorm as follows:

$$norm(\mathbf{x}^*) = \mathbf{J_n^+}(\mathbf{x}^*) \cdot \mathbf{x}^* \tag{6}$$

At inference for a given input, we now have a linear RMSNorm whose output is numerically identical to the one used in training. However, when we take the gradient with respect to the input vector $\mathbf{x}$ in *eval* mode, the numerical output is the detached Jacobian matrix $\mathbf{J_n^+}$, which we can use to reconstruct the normalization output as a linear system.

The goal is to apply this same approach for other nonlinear functions in the decoder such that the entire computation from the input embedding vectors to the predicted output is linear for a given input, and we can compute and interpret the set of detached Jacobian matrices.

### 2.3.2 Activation functions

While Mohan et al. (2019) relied on *ReLU* activation functions, which do not require any changes to achieve linearity, Llama 3.2 3B uses SwiGLU (Shazeer, 2020), Gemma 3 uses approximate GELU (Hendrycks & Gimpel, 2016) and Qwen 3 uses Swish for activation functions. There is a linear $\mathbf{x}$ term in each of these, and the gradients can be cloned and detached from the nonlinear terms. This manipulation produces a pointwise linear Swish layer with respect to the input $\mathbf{x}$.

$$\text{Swish}\,(\mathbf{x}) = \mathbf{x} \cdot \text{sigmoid}(\beta \cdot \mathbf{x}) \tag{7}$$

$$\text{Swish}\,(\mathbf{x}^*) = \mathbf{x} \cdot \text{sigmoid}(\beta \cdot \mathbf{x})|_{\mathbf{x}=\mathbf{x}^*} \tag{8}$$

$$\text{Swish}\,(\mathbf{x}^*) = ([\frac{\partial}{\partial \mathbf{x}}\text{Swish}(\mathbf{x})]|_{\mathbf{x}=\mathbf{x}^*}) \cdot \mathbf{x}^* \tag{9}$$

$$\text{Swish}\,(\mathbf{x}^*) = \mathbf{J_{Swish}^+}(\mathbf{x}^*) \cdot \mathbf{x}^* \tag{10}$$

Detaching the gradient from the Swish output thus allows for a pointwise linear form of Swish at inference. A similar procedure may be carried out for SwiGLU with Llama 3 and GELU with Gemma 3 (see supplement, eq. 17).

### 2.3.3 Attention

The softmax operation at the output of the attention block can also be detached, with the linear relationship preserved through the subsequent multiplication with $\mathbf{V}$, which is a linear function of $\mathbf{x}$. Below, $\mathbf{Q} = \mathbf{W_Q}\mathbf{x}$, $\mathbf{K} = \mathbf{W_K}\mathbf{x}$ and $\mathbf{V} = \mathbf{W_V}\mathbf{x}$.

$$Attn(\mathbf{Q}, \mathbf{K}, \mathbf{V}) = softmax(\frac{\mathbf{Q}\mathbf{K}^T}{\sqrt{d_k}}) \cdot \mathbf{V} \tag{11}$$

$$Attn(\mathbf{x}) = [softmax(\frac{\mathbf{Q}\mathbf{K}^T}{\sqrt{d_k}})|_{\mathbf{Q}=\mathbf{Q}^*,\mathbf{K}=\mathbf{K}^*}] \cdot \mathbf{W_V}\mathbf{x} \tag{12}$$

$$Attn(\mathbf{x}^*) = ([\frac{\partial}{\partial \mathbf{x}}Attn(x)]|_{\mathbf{x}=\mathbf{x}^*}) \cdot \mathbf{x}^* \tag{13}$$

$$Attn(\mathbf{x}^*) = \mathbf{J_{Attn}^+}(\mathbf{x}^*) \cdot \mathbf{x}^* \tag{14}$$

The linear $\mathbf{x}$ term within $\mathbf{V}$ makes it possible for the attention block to be pointwise linear at inference, as the gradient for the softmax output is detached.

### 2.3.4 The Transformer Decoder

With the the above gradient detachments for the normalization layers, activation functions and attention blocks, the transformer decoder network is linear with respect to $\mathbf{x}^*$ when evaluated at $\mathbf{x}^*$ (shown here for length $k$).

$$\mathbf{y}^* = \sum_{i=0}^{k} \mathbf{J}_\mathbf{i}^+(\mathbf{x}^*) \cdot \mathbf{x}_\mathbf{i}^* \tag{15}$$

The output of the network incorporating the above gradient detachments is unchanged from the original architecture but has an equivalent linear representation.

## 3 Results

### 3.1 Pointwise linearity of the predicted output

In order to validate whether the detached Jacobian achieves reconstruction with a linear representation, we can compare the predicted output embedding vector for a given input token sequence to the reconstruction of the output. As a baseline, we can also compute the reconstruction using the conventional Jacobian as in Mohan et al. (2019) and examine its accuracy. Given the above argument that the appropriate gradient detachments are necessary to achieve output reconstruction, we expect the detached Jacobian to accomplish reconstruction, but the conventional Jacobian to fail.

Fig. 1B compares the network output to both the conventional and detached Jacobian reconstructions for Llama 3.2 3B and Qwen 3 14B. The reconstruction of the output embedding with the detached Jacobian matrices falls on the identity line when compared with the output embedding, showing accurate reconstruction, while the reconstruction with the conventional Jacobian is not at all close to the output. This comparison therefore demonstrates the validity of the reconstruction with the linear system of the detached Jacobian for Qwen 3 14B for a particular input.

In order to examine the fidelity of the detached Jacobian reconstruction, we compared the reconstruction against the network output using PyTorch function allclose with varying tolerance levels. The reconstructions achieved numerical agreement within a relative and absolute tolerance of $10^{-13}$. This tolerance is approximately 50 times the machine epsilon of $2.2 \cdot 10^{-16}$ for 64-bit floating-point numbers, indicating high-fidelity reconstruction that is numerically equivalent to the reference implementation for practical purposes. As an additional metric, the norm of the detached Jacobian reconstruction error divided by the norm out of the output is on the order of $10^{-14}$.

The numerical computation of the full detached Jacobian matrix takes on the order of 10 seconds for an input sequence of 8 tokens for Llama 3.2 3B in float32 on a GPU with 24 GB VRAM. In contrast, the full Jacobian matrix for the same sequence at float64 precision with Qwen 3 14B on a GPU with 40 GB VRAM takes 20 seconds. An approximate method for computing the top $k$ singular vectors of the detached Jacobian without forming the full matrix utilizing Lanczos iteration has also been implemented in JAX for Gemma 3 4B, allowing for the efficient computation of the top 16 singular vectors of the detached Jacobian for up to 100 input-token input. The maximum length tested on a GPU with 80 GB VRAM was over 400 tokens for only the top singular vector corresponding to each token. The Lanczos method trades reconstruction precision for scalability while preserving interpretability, and examples are available in the code repository.

### 3.2 Single-unit feature selectivity and invariance

Since the detached Jacobian applied to the input embedding reproduces the predicted output embedding vector, and the elements of the predicted output embedding vector are the units of the last transformer layer, the rows of the detached Jacobian matrices represent the input features to which the last layer units are selective and invariant for that particular input sequence (Kadkhodaie et al., 2023; Mohan et al., 2019).

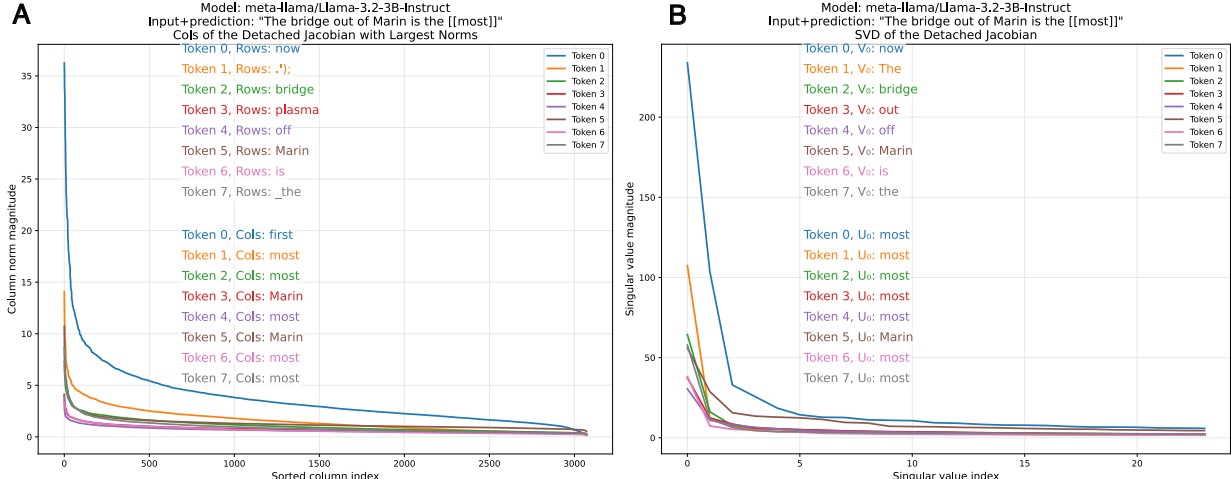

Figure 2: Given the sequence "The bridge out of Marin is the", the most likely prediction is "most" for Llama 3.2 3B. The detached Jacobian matrices for each token represent an equivalent linear system that computes the predicted output embedding. A) We show the features which drive large responses in single units in the last decoder layer, which are the rows of the detached Jacobian with the largest norm values, and decode each of those into the most likely input embedding token. The block of words at the top shows the ordered decoded "feature" input tokens from the largest rows of the detached Jacobian matrix for the input tokens. A similar operation is carried out for columns of the largest norm values, which are decoded to the output token space. Note that the activation distribution of column magnitudes is fairly sparse, with only a few units driving the response. B) We take the singular value decomposition of the detached Jacobian matrix corresponding to each input token, which summarizes the modes driving the response, and decode the right and left singular vectors $V$ and $U$ to input and output embeddings, shown in colors. The singular value spectrum is extremely low rank, and decoding the $U$ singular vectors returns candidate output token, including "most" and "first". Decoding the $V$ singular vectors returns variants of the input tokens like "bridge", "Marin" and "is", as well as others that are not clearly related to the input sequence.

The activation of a particular unit in the last layer is determined by the inner product of a row of the detached Jacobian and the input embedding vector. We can sort by the magnitude of row norms, then map the largest-magnitude rows of the detached Jacobian back to the input embedding space (via cosine similarity to the input embedding matrix, since input embeddings are not typically mapped back to tokens during normal model operation) to determine the tokens that cause each unit to be strongly positive or negative. We can see in Fig. 2A that the units respond strongly to the words of the prompt, including "bridge", "Marin" and "is". Decoding of the rows of the detached Jacobian for each token as well as the distribution of activations for this sequence is shown in Fig. 2A. The columns of the Jacobian can also be decoded in the conventional manner to the output token space with the unembedding layer, and these turn out to be tokens that could be predicted, which include words like "most" or "first", which could be acceptable outputs.

### 3.3 Singular vectors of the detached Jacobian

An alternative approach is to look at the singular value decomposition of the detached Jacobian $\mathbf{J_i^+ = U\Sigma V^T}$, following Mohan et al. (2019). Since the detached Jacobian represents the forward computation, the fact that the SVD is very low rank shows the entire forward computation can be approximated with only a few singular vectors operating on the input embeddings.

Unlike image denoising models (Mohan et al., 2019; Kadkhodaie et al., 2023) where input and output spaces are similar and singular vectors U and V have a high cosine similarity, corresponding left and right singular vectors of LLMs differ substantially. This reflects the asymmetric nature of next-token prediction, as right

singular vectors V capture which input token features drive the computation, while left singular vectors U capture which output token directions are predicted.

In Fig. 2B, the singular vectors are decoded for Llama 3.2 3B (and for other models in supplemental Fig. A3). The right singular vectors $V$ are decoded to input tokens in the same way the rows of the detached Jacobian were above (nearest-neighbor to input embeddings from cosine similarity), and we see similar decoding of the top tokens to the features driving the most active single units. The left singular vectors $U$ can be decoded to output embedding tokens (with the conventional method from the unembedding matrix), and "most" is the strongest, as it was in the columns of the detached Jacobian matrices.

| | Input token 0 | Input token 1 | Input token 2 |
|---|---|---|---|
| Layer 25_0 | largest most first longest latest fastest last third | bridge bridges Bridge gateway | hardest ones exit easiest first most fastest highway |
| Layer 25_1 | bridge bridges Bridge Bridges brid | bridges bridge Bridge bridge Bridge | ( exit exit exits eternity . exit |
| Layer 25_2 | bridges bridge Bridge bridge parliament | Exit exit jams | INCIDENT symbolism |
| Layer 26_0 | first most largest last longest latest gateway only | bridge bridges metaphor gateway connecting | highway first exit ones last hardest roads |
| Layer 26_1 | bridge bridges metaphor Bridges Bridge | bridges bridge structures brid bridge | .charset jams Margins |
| Layer 26_2 | parliament structures bridges Parliament bridge | Exit exit choke Exit panicked | symbolism metaphor |
| Layer 27_0 | first last largest bridge longest most oldest latest | bridge bridges Bridge Bridges | last first exit highway bottleneck next road choke |
| Layer 27_1 | bridge bridges Bridge Bridges | bridges bridge Bridge bridge brid Bridge | EXIT exit exits ( exit |
| Layer 27_2 | bridge bridge bridges Bridge structures | Exit exit Exit | exit . exit incident EXTRA incidents |
| Layer 28_0 | bridge longest largest first busiest last oldest most | bridge bridges Bridge Bridge | highway exit bottleneck highways Highway last road exits |
| Layer 28_1 | bridge bridges Bridge Bridge | highway highways coast freeway roads road route | exit exits EXIT exit Exit |
| Layer 28_2 | bridge bridge bridges Bridge brid | Exit exit Exit exit exit | exit Saddam Mosul Kuwait incident metaphor |
| Layer 29_0 | bridge only fourth last third longest fifth most | bridge bridges Bridge Bridges | only last first highway third highways exit fourth |
| Layer 29_1 | bridge bridges Bridge Bridges | coast highway road driveway coastline roads highways freeway | exits exit EXIT |
| Layer 29_2 | bridge bridges bridge structures brid structure | Exit exit Highway Exit | Saddam Mosul Elvis metaphor incident |
| Layer 30_0 | bridge most longest fourth third last only fifth | bridge bridges Bridge Bridge | highway only bridge last first Highway road highways |
| Layer 30_1 | bridge bridges Bridges Bridge | coast freeway highway coastline road roads highways | bridge Bridge bridges bridge brid |
| Layer 30_2 | bridge structure structures bridges bridge brid | sail seab sailing Bermuda ship | Memphis Kuwait Jordan Saddam Iowa |
| Layer 31_0 | bridge most only last longest first third largest | bridge bridges Bridge Bridge | only last highway first bridge exit Highway most |
| Layer 31_1 | coast airlines Interior airline interior Lua Speedway | coast coastline coastal Coast Coastal route | bridge Bridge bridges bridge underwater brid |
| Layer 31_2 | bridge bridges bridge brid Bridge structure | ship sail sailing dock seab | Jordan Memphis Kuwait Mississippi |
| Layer 32_0 | bridge most only first last longest third largest | bridge Bridge bridges Bridge bridge | only last first highway most main route exit |
| Layer 32_1 | interior airline steam airlines Trail breed vacuum | coast coastal coastline route Coast Route beach | bridge span underwater connecting deck public member |
| Layer 32_2 | bridge bridge bridges Bridge brid Bridge | ship sail dock sailing seab | Kuwait Jordan Memphis Edmonton Mississippi Nile |
| Layer 33_0 | only first last most third main second subject | bridge Bridge bridges Bridge only | only last first key main same most exit |
| Layer 33_1 | planet interior cabin floors roots | coast coastline coastal Coast route beach Coastal | span public member library platform floating intervening deck |
| Layer 33_2 | bridge bridge structure bridges brid Bridge | ship orbit aircraft sail vessel | Kuwait Nile Edmonton Saskatchewan Tulsa |

Table 1: The top eight tokens decoded from the largest three singular vectors of the detached Jacobian for the layer outputs from Qwen 3 14B for the sequence "The bridge out of Marin is the" with the prediction [[only]]. Legend: "Bridge", "only", "highway", "exit", "most". Semantic concepts emerge clearly by layer 25. The predicted token 'only' appears prominently in later layers alongside related infrastructure and geographic concepts. Note the progression from general bridge concepts in early layers to specific architectural terms (span, deck, platform, floating), geographic terms (coast, coastline, route, beach) and locations with notable bridges in the final layer. See also supplemental Tables 3, 4 and 5 for the longer tables for Llama, Gemma and Qwen.

## 3.4 Comparative Analysis of Singular Vectors in Llama 3 and Qwen 3

A direct comparative analysis of the singular vectors derived from the detached Jacobian matrices of Llama 3 3.2B and Qwen 3 4B offers a lens through which to view not only the shared computational principles of modern LLMs but also their distinct data-driven approaches. While both models demonstrate a consistent hierarchical organization of their predictive computations, they diverge significantly in their semantic richness, their approach to multi-lingual representations, and their tokenization strategies. These differences are made visible by the SVD of their equivalent linear mappings and reveal unique styles that likely reflect their underlying training datasets.

In terms of their singular value spectra over 100 examples, Fig. 3 shows that both Llama 3 and Qwen 3 are consistently low-rank. The first token for Qwen 3 has a low average rank at 1.01 than Llama 3 at 1.06, but Qwen's next singular vectors are all higher rank than those of Llama. Llama's "beginning of text" token is surprisingly of lower rank than the first text token.

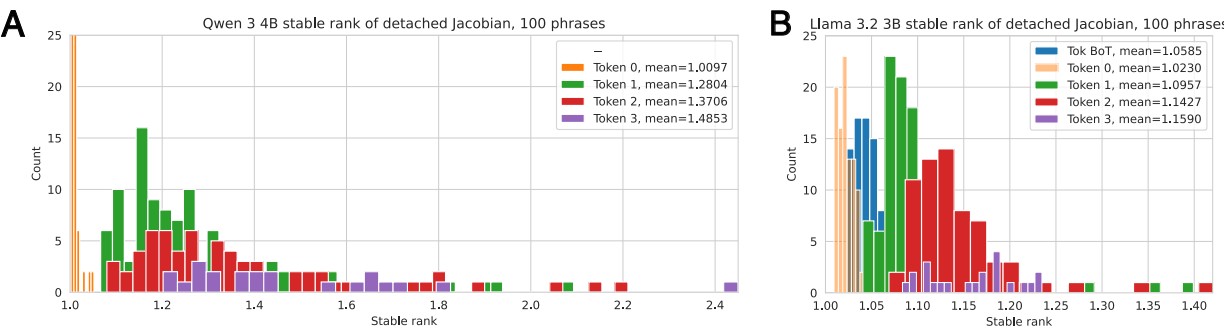

Figure 3: For 100 short input phrases, the stable rank distribution as a function of input token number. Note that Llama 3.2 3B uses a $<|BoT|>$ token and Qwen 3 4B does not.

In terms of the semantic content of the singular vectors, both Llama 3 and Qwen 3 employ a similar hierarchical strategy. The first singular vector $U_0$ with largest magnitude establishes the foundational layer of prediction. This vector primarily contains high-frequency tokens that provide grammatical structure or represent the most probable continuations. For example, in "Should have known," both models place "better" and common punctuation in their $U_0$ vectors. This shared pattern reinforces the hypothesis that the dominant computational axis in transformers is dedicated to establishing a coherent structural and high-probability scaffold upon which more nuanced semantic meaning can be built. See section A.5 in appendix for more examples of each of these analyses. **21 phrases out of 100** fit this category.

### Llama 3 (Abstract Semantics) vs. Qwen 3 (Direct Semantics)

A distinction in semantic processing is pronounced in the secondary singular vectors ($U_1$ and $U_2$). Llama 3 consistently demonstrates a rich and abstract English-centric semantic space. For the input "Will break," its $U_1$ vector contains a diverse set of conceptual possibilities like "confidentiality," "independence," "promises," and "ground." This indicates a capacity to reason about abstract concepts that can be "broken." Qwen 3's vectors for the same phrase are more direct and action-oriented, featuring tokens like "ties," "neck," and "dance," alongside Chinese characters for "stiff" and "can't." This highlights Llama 3's deep modeling of the nuances and abstractions within the English language. **14 phrases out of 100** fit this category.

### Qwen 3's Multilingual Reasoning

Perhaps the most obvious difference revealed by this analysis is Qwen 3's multilingual and cross-lingual representation capability, which is largely absent in Llama 3's vectors for the analyzed English prompts. In nearly every example, Qwen 3's secondary vectors are populated with non-English tokens—primarily Chinese, but also Russian and others that are conceptually related to the input phrase. For "The broken," Qwen 3's $U_1$ vector includes Chinese tokens for "bicycle," "vase," "necklace," and "window"—all concrete examples of breakable objects. This demonstrates that Qwen 3 does not operate in a constrained linguistic space; rather, it accesses a unified, cross-lingual conceptual representation to generate predictions. **38 phrases out of 100** fit this category.

### Examples of sub-word Fragments in Qwen 3

We also observed a difference in tokenization and morphological strategy. Qwen 3's secondary vectors frequently contain what appear to be sub-word fragments or tokenization artifacts (e.g., "e," "eer," "ection," "ing"). The persistent recurrence of these tokens, often in the $U_2$ vector, suggests that part of Qwen 3's computational process involves constructing or modifying words at a morphological level. This could be an efficient mechanism for handling its multilingual vocabulary. Llama 3 tends to operate with whole-word semantic tokens, indicating a different approach to vocabulary representation. **33 phrases out of 100** fit this category.

### 3.5 Layer output singular vectors

Table 1 shows the top eight tokens decoded from the largest three singular vectors of the detached Jacobians of selected layer outputs for Qwen 3 14B. The words "bridge" (and its variants), "highway", "exit", "most" and "only" are highlighted to show their appearances in decoded singular vectors. Early layers are excluded as the tokens are unintelligible. The emergence of intelligible tokens in later layers is shown in the tables as something like a phase change in the representation. Qwen 3 generates infrastructure and engineering related concepts before producing "only".

Fig. 4A shows the normalized singular value spectra of the detached Jacobian at the output of every layer. Llama 3.2 3B has 28 layers, and decoding the largest singular vectors shows that the word representation of these intermediate operations is not interpretable until later layers. From the decoding of the top singular vector by layer, "only" emerges in layer 19. From the map of the progression of the projection of the top two singular vectors onto the top two singular vectors of the last layer in Fig. 4B, we first see a shift at layer 11 toward the prediction.

Since the layer-by-layer operations are only linear, the stable rank $R = (\sum_i^L S_i^2)/S_{max}^2$ serves as a measure of the effectively dimensionality of the subspace of the representation at a particular layer.

When looking at $W_{0\_to\_k}$, the cumulative layer transform up through layer $k$, the dimensionality of the detached Jacobian steadily decreases. When considering each layer $i$ as its own individual transform $W_i$ (where $W_{0\_to\_k} = \prod_{i=0}^k W_i$ for the simplified scenario of a single input token; there are other cross-token terms not shown here for mid-layer detached Jacobians for longer input sequences), we also see a large peak in dimensionality near the end.

| Model | Layer intervention | Input sequence | Normal response | Steered response |
|---|---|---|---|---|
| Llama 3.1 8B | 24 / 36 | 'I'm going to arizona to see the' | 'I'm going to arizona to see the Grand Canyon. I've heard it's a must see. I've also heard it's a bit of a trek to' | 'I'm going to arizona to see the Grand Canyon, and I'm planning to hike the Bright Golden Gate Bridge (I think that's the name of the trail) in the Grand Canyon.' |
| Qwen 3 8B | 24 / 36 | 'Here is a painting of the' | 'Here is a painting of the same scene as in the previous question, but now the two people are standing on the same side of the building. ' | 'Here is a painting of the Golden Gate Bridge in San Francisco. The Golden Gate Bridge is one of the most famous bridges in the world. ' |
| Gemma 3 12B | 33 / 48 | 'I went to new york to see the' | 'I went to new york to see the memorial and museum. It was a very moving and emotional experience.' | 'I went to new york to see the 10th anniversary of the Broadway show, "The Golden Gate Bridge Bridge." It was a great show.' |

Table 2: Detached Jacobian matrices as steering operators, pilot results with Llama 3.1 8B, Qwen 3 8B and Gemma 3 12B.

### 3.6 The detached Jacobian as a conceptual steering operator

Steering vectors are a well-known technique for altering LLM outputs (Liu et al., 2023) where a vector with certain properties is added to a mid-layer representation, and the sum is passed through the rest of

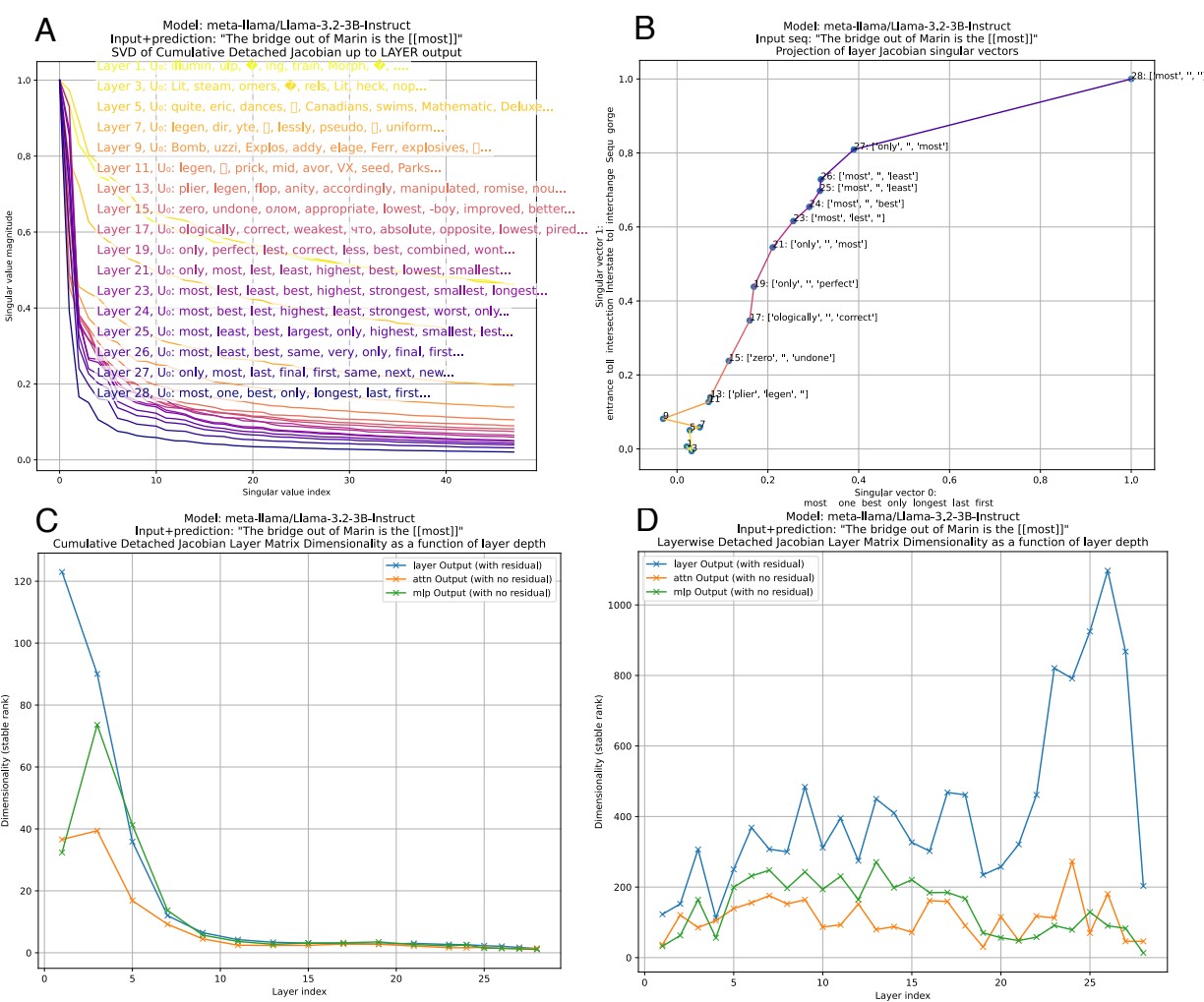

Figure 4: Since the transform representing the model forward operation is linear after detachment, we can also decompose each transformer layer as a linear operation as well. A) The singular value spectrum for the cumulative transform up to layer $i$. Note that later layers are lower rank than earlier layers. The top singular vectors of the later layers show a clear relation to the prediction of "most". B) The projection of the top two singular vectors onto the top two singular vectors of the final layer. The singular vectors of the first 10 layers are very different than those of the last layer, so the projections remain close to the origin. At layer 11, they begin to approach those of the output layer. C) A measurement of the dimensionality of the cumulative transform up to the output of each layer as the stable rank. Within each layer, the outputs of the attention and MLP modules (prior to adding the residual terms) can also be decomposed as linear mappings. The dimensionality decreases deeper into the network at each of these points, except for a slight increase for the attention and MLP module outputs in layer 3. D) The dimensionality of the detached Jacobian for the layer-wise transform at layer $i$ for the layer output, as well as the attention module output and MLP module output.

the network to generate an output token. Here we utilize the detached Jacobian as an operator instead of an additive vector, and compute it from an intermediate layer for a "steering" phrase like "The Golden Gate" (after the "Golden Gate Claude" demo (Templeton et al., 2024)). The model predicts "Bridge", and this detached Jacobian matrix is used to steer the continuation of a new phrase toward this concept. For a new input phrase, like "Here is a painting of the", the "new" input sequence's embedding vectors $\mathbf{x}^*_{\mathbf{new}}$ are multiplied by the detached Jacobian previously computed from the steering concept $\mathbf{J}^+_{\mathbf{L}}(\mathbf{x}^*_{\mathbf{steer}})$, scaled by $\lambda$ and added to the layer activation $\mathbf{f}_{\mathbf{Li}}$ from the "new" input.

$$\mathbf{f_{Li}(x)} = \lambda \cdot \mathbf{f_{Li}(x^*_{new})} + (1 - \lambda) \cdot \mathbf{J^+_{Li}(x^*_{steer})} \cdot \mathbf{x^*_{new}} \tag{16}$$

This steered intermediate representation is then put through the remaining layers of the network and the next token is decoded. The detached Jacobian must only be computed once for the steering concept, and therefore this method is rather efficient. Table 2 shows how the detached Jacobian from an intermediate layer imposes the Golden Gate Bridge as the semantic output coherent with the rest of the input sentence, even when it is difficult to make a logical connection. Beyond demonstrating practical utility, the success of the steering operator provides validation that the detached Jacobian captures actual semantic representations.

## 4 Discussion

The detached Jacobian approach allows for linear representations of the transformer decoder to be found for each input sequence, without changing the output. The intermediate outputs of each layer and the attention and MLP modules are also accurately reproduced by the detached Jacobian function.

The detached Jacobian operation is accurate only at the specific operating point at which the matrices were computed by the PyTorch autograd function. A short distance away in the input embedding neighborhood, the detached Jacobian will be extremely different because the manifold is highly curved. (Although local neighborhood validity is less applicable to LLMs which map tokens to embedding vectors, as inputs will only ever discretely sample the embedding space, and there is not an obvious need for exploring the local neighborhood to embedding vectors that do not represent words from the input vocabulary). The manifold is not piecewise linear, but only has a linear equivalent at the operating point, which can be found numerically for every input sequence.

## 5 Conclusion

While our current analysis covers a limited range of examples, the approach suggests a path toward large-scale interpretability by computing the detached Jacobian for many token predictions in a given dataset and analyzing the resulting linear systems to understand semantic patterns across diverse contexts. Given the low-rank nature of the detached Jacobian, our Lanczos method, which efficiently computes only the top singular vectors of the Jacobian, is a step toward making this practical. Future work should explore this scaling potential, moving toward comprehensive equivalent linear analysis of LLM behavior across tasks, domains, and model architectures.

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

# A  Appendix

## A.1  Code availability

Code is provided as a zip file (and will be made available on github).

## A.2  Pointwise linear GELU

Gemma 3 uses the approximate GELU activation function. Below $\gamma = 0.44715$. Here is the derivation of the pointwise linear version of GELU used for Gemma 3 in the preceding analysis.

$$\mathrm{GELU}\left(\mathbf{x}\right) = \frac{1}{2}\mathbf{x}\left(1 + \tanh\left[\sqrt{2/\pi}\left(x + \gamma\mathbf{x}^3\right)\right]\right) \tag{17}$$

$$\mathrm{GELU}\left(\mathbf{x}\right) = \frac{1}{2}\mathbf{x}\left(1 + \tanh\left[\sqrt{2/\pi}\left(x + \gamma\mathbf{x}^3\right)\right]\right)|_{\mathbf{x}=\mathbf{x}^*} \tag{18}$$

$$\mathrm{GELU}\left(\mathbf{x}^*\right) = \left([\frac{\partial}{\partial\mathbf{x}}\mathrm{GELU}(x)]|_{\mathbf{x}=\mathbf{x}^*}\right)\cdot\mathbf{x}^* \tag{19}$$

## A.3  Singular vectors across model families

Fig A3 shows this same analysis for Llama 3, Qwen 3 and Gemma 3 across two different sizes of each. Note the low-rank structure of each of the detached Jacobians, as well as the differing decoding of the top singular vectors from each input embedding vector. The first or "beginning of sequence" token has the highest magnitude in each spectrum reflecting how the positional encoding is entangled with semantic information in the detach Jacobian representation.

## A.4  Additional models

Pointwise linearity for Deepseek R1 0528 Qwen 3 8B Distill, Phi 4, Mistral Ministral and OLMo 2 are shown on the following page. See Fig. A4.

## A.5  Examples for comparative analysis of singular vectors in Llama 3 and Qwen 3

**Shared High-Probability Tokens in $U_0$**

This pattern shows both models using their primary singular vector ($U_0$) to establish a foundation of common, structurally likely next words.

For the phrase "To see," both models prioritize articles and question words.

- **Qwen 3** $U_0$: the a this an all how what and
- **Llama 3** $U_0$: the a , and what an if

For "To complete," both models identify determiners as the most probable continuations.

- **Qwen 3** $U_0$: the a this his an my your
- **Llama 3** $U_0$: this , the a and an (

For "The final result," the $U_0$ vectors in both models are dominated by common prepositions and linking verbs that would grammatically follow the phrase.

- **Qwen 3** $U_0$: of is in for from after , was

- **Llama 3** $U_0$: `, of ... ( is in and`

Both models use their primary singular vector ($U_0$) to propose very similar sets of common, structurally-likely next words. This highlights a shared foundational strategy of prioritizing grammatical coherence.

**21 phrases out of 100** fit this category.

- **Before they:** Both suggest verbs like `were, can, could, start`.

- **While walking:** Both suggest prepositions of movement like `in, through, on, along, around`.

- **To see:** Both prioritize articles (`the, a`) and question words (`what, how`).

- **Will break:** Both suggest particles like `down` and `up`, and articles like `the, a`.

- **Must leave:** Both list determiners (`the, a, this`) and prepositions (`in, at`).

- **Should take:** Both include `a, the, into`, and `care`.

- **After reading:** Both list `the, this, a, about`, and `"`.

- **When finished:** Both suggest `,` and `with`.

- **To begin:** Both prioritize `,` and `with`.

- **May open:** Both suggest `a, the, up`, and `in`.

- **Could drive:** Both include `a, the, in`, and `,`.

- **During lunch:** Both list `„ time`, and `break`.

- **To learn:** Both prioritize `the, more, about`, and `how`.

- **The green:** Both include `and, „ light, is`.

- **The old man:** Both list linking verbs (`was, is`) and conjunctions (`and`).

- **To build they:** Both suggest modal verbs (`have, need, must, would`).

- **The fast car:** Both include `is, has, and, ,`.

- **The tall building:** Both list `is, in, has, with`.

- **To create:** Both prioritize articles `a, an, the`.

- **The response:** Both include `to, is, of`.

- **The solution:** Both list `to, of, is, for`.

### Llama 3 (Abstract Semantics) vs. Qwen 3 (Direct Semantics)

This pattern illustrates how Llama 3's secondary vectors often explore a wider and more abstract conceptual space compared to Qwen 3's more direct and action-oriented suggestions.

For the phrase "Should take," Llama 3 suggests abstract responsibilities or concepts one should "take on," while Qwen 3 suggests direct objects or actions.

- **Llama 3** $U_1$: `utmost admission inspiration revision discipline quitting responsibility guidance`

- **Qwen 3** $U_1$: `refuge aways -away brib` 半天 (half-day) 午饭 (lunch) `away` 这笔 (this sum)

For "To imagine," Llama 3's vectors include abstract and philosophical concepts to imagine, whereas Qwen 3 focuses on more concrete items like "scenarios."

- **Llama 3** $U_1$: `reconstruct ethical erect owning peace embodied meanings yourself`

- **Qwen 3** $U_1$: `scenarios` 场景 (scene) `scenario` 也是一种 (is a kind of) `oha Scenario worlds Scenario`

For "The discovery," Llama 3's vectors describe the impact and nature of a discovery (revolutionary, baffling), while Qwen 3's vectors describe the event of a discovery (a journey, an unintentional bulletin).

- **Llama 3** $U_1$: `revolution shed bust of details vind catapult baff`

- **Qwen 3** $U_1$: 震惊 (shock) 轶事 (anecdote) 新西 (New West/ New Zealand) 了一个 (a) 之旅 (journey) 无意 (unintentional) 快报 (bulletin) 小镇 (small town)

### A.5.1 Llama 3 (Abstract Semantics) vs. Qwen 3 (Direct Semantics)

Here, Llama 3's secondary vectors explore broader, more abstract concepts, while Qwen 3's are more concrete and action-oriented.

**14 phrases out of 100** show this strong contrast.

- **Will break:** Llama `confidentiality, independence`; Qwen `ties, neck, dance`.

- **Must leave:** Llama `departing, orientation`; Qwen `immediately, room`.

- **Should take:** Llama `admission, inspiration, discipline`; Qwen `refuge, advantage`.

- **The broken:** Llama `fragments, promises, torn`; Qwen `window, clock, vase`.

- **To begin:** Llama `brainstorm, conceptual`; Qwen `start, validate`.

- **May open:** Llama `invitation, plea`; Qwen `windows, sesame`.

- **Could drive:** Llama `distracted, fleets, uninsured`; Qwen `drunk, uphill`.

- **The discovery:** Llama `revolution, catapult`; Qwen `journey, bulletin`.

- **To prevent:** Llama `vulnerability, security`; Qwen `corrosion, fires`.

- **The solution:** Llama `vector, lattice, eigen`; Qwen `set, definition`.

- **To complete:** Llama `projects, tasks`; Qwen `orders, assignment`.

- **Were planning:** Llama `launching, upcoming`; Qwen `permission, meetings`.

- **The evidence:** Llama `overwhelmingly, against`; Qwen `suggests, linking`.

- **To create:** Llama `customized, empowering`; Qwen `custom, interactive`.

### Qwen 3's Multilingual Reasoning

This pattern showcases Qwen 3's unique ability to access a cross-lingual conceptual space, populating its secondary vectors with semantically relevant non-English tokens.

For the phrase "The fast car," Qwen 3's $U_1$ vector includes multiple Chinese words related to speed and motion.

- **Qwen 3** $U_1$: `overt` 的速度 (speed) 运动 (motion) `.Speed` 追赶 (chase) 速度 (speed) `riages` 超越 (surpass)

For "The fresh bread smelled," Qwen 3's $U_2$ vector is a list of Chinese synonyms and related concepts for "smell" and "fragrance."

- **Qwen 3** $U_2$: `smell smells` 嗅 (sniff/smell) 闻 (smell/hear) 香 (fragrant) 香气 (aroma/fragrance) 香味 (fragrance/scent) `smelling`

For "Should help her," the $U_1$ vector remarkably contains relevant concepts from multiple languages, including Chinese (career development, alleviate), Russian (cope/handle), and Vietnamese (support/help).

- **Qwen 3** $U_1$: 事业发展 (career development) справиться (handle/cope) hỗ trợ (support/help) 缓解 (alleviate) `unpack` 管理工作 (manage work) 过渡 (transition) 学业 (studies)

**38 phrases out of 100** contain clear examples of multilingual reasoning.

**Examples of Sub-word Fragments in Qwen 3**

For the phrase "While walking," the second singular vector for the token "walking" is almost entirely composed of these fragments, including common suffixes.

- **Vector (Token 1, $U_2$)**: `e ection ing eer` ignKey `cion ging eed`

For "To begin," the $U_2$ vector includes the common suffixes -ments and -ly, suggesting a mode for building nouns and adverbs.

- **Vector (Token 1, $U_2$)**: `e ments eel hips eed s eve ly`

For "The deep water," the $U_2$ vector for the token "water" contains fragments like -ness and -ection.

- **Vector (Token 2, $U_2$)**: `e y eer eel eus ness ection yth`

### A.5.2 Qwen 3's "Word-Building" Vector

This category identifies phrases where a secondary Qwen 3 vector is dominated by sub-word fragments and morphological units (e.g., -ing, -tion, -eer, -ness).

**33 phrases out of 100** clearly display a dedicated morphological vector.

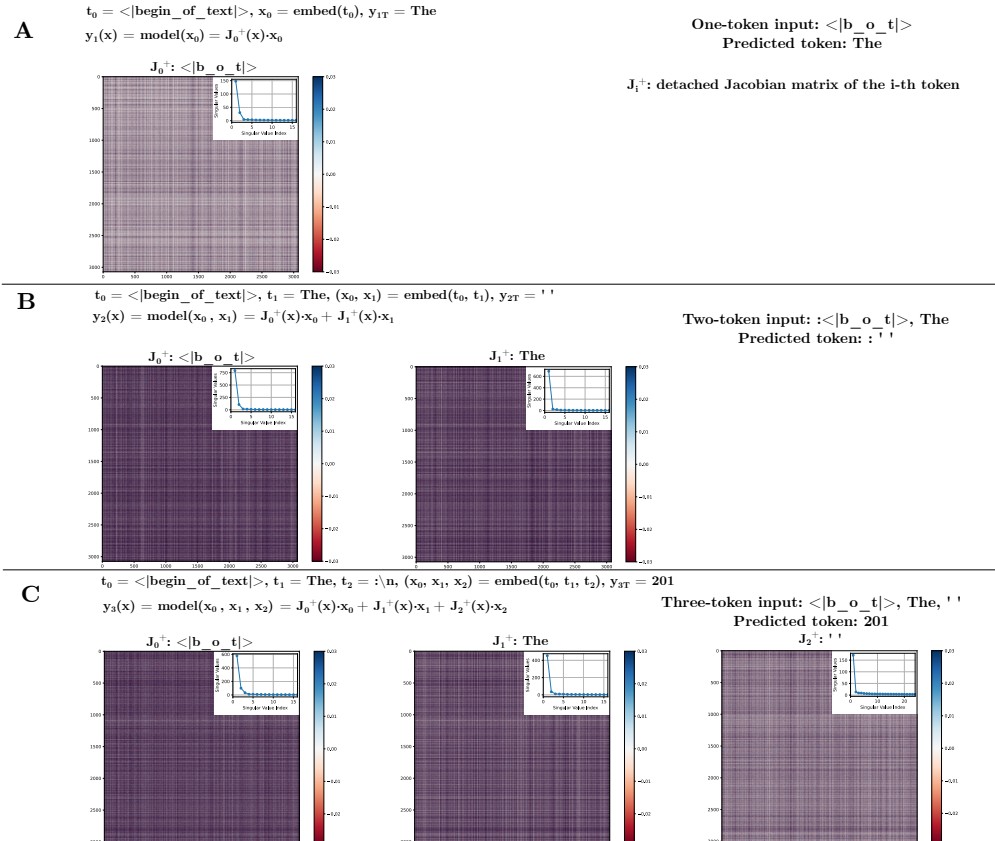

Figure A1: An overview of next-token prediction in the Llama 3.2 3B transformer decoder and decomposition of the predicted embedding vector computation using the detached Jacobian. Generating three tokens with only $< |BoT| >$ as input produces "The 201". For each prediction, each input token $\mathbf{t_i}$ is mapped to an embedding vector $\mathbf{x_i}$, and the network generates the embedding of a next token. The phrase turns out to be "The 2019-2020 season". The detached Jacobian $\mathbf{J}^+(\mathbf{x})$ of the predicted output embedding with respect to the input embeddings is composed of a matrix corresponding to each input vector. Each detached Jacobian matrix $\mathbf{J_i^+}(\mathbf{x})$ is a function of the entire input sequence but operates only on its corresponding input embedding vector. The matrices tend to be extremely low rank, shown in the inset figures, and the matrix $\mathbf{J_0^+}$ varies across A), B) and C) above because the input sequences differ. Since the detached Jacobian captures the entirety of the model operation in a linear system (numerically, for a given input sequence), tools like the SVD can be used to interpret the model and its sub-components.

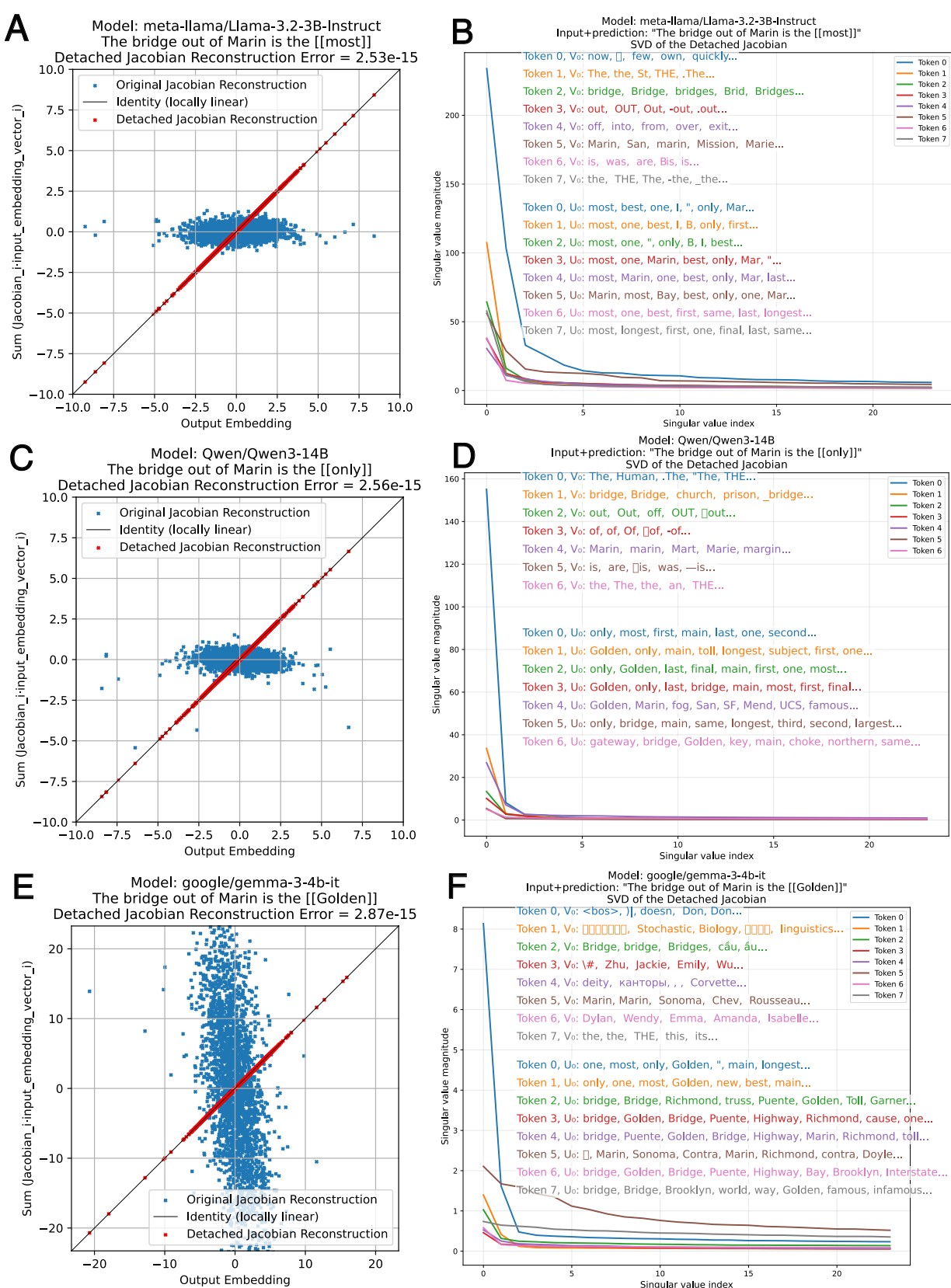

Figure A2: The detached Jacobian reconstruction error and SVD for Llama 3.2 3B, Qwen 3 14B and Gemma 3 4B

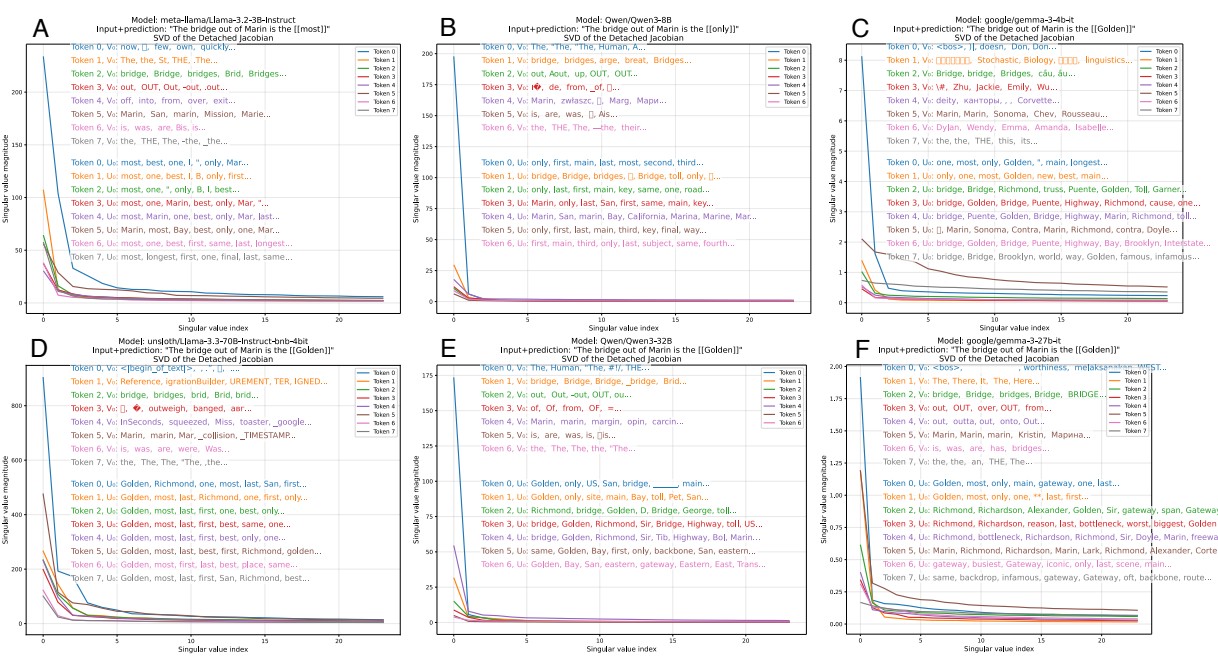

Figure A3: Singular value decomposition of the detached Jacobian for different families and sizes of language models (from 3B to 70B parameters) evaluating the input sequence "The bridge out of Marin is the", followed by a predicted token. The left singular vectors decode to tokens related to bridges and local geography, particularly the Golden Gate Bridge, while singular value spectra all have extremely low rank (see below for quantification). Each row shows top tokens associated with different singular vectors, demonstrating how models encode semantic knowledge about the input sequence and the prediction. See Fig. A4 for Deepseek R1 0528 Qwen 3 8B Distill, Phi 4, Mistral Ministral and OLMo 2.

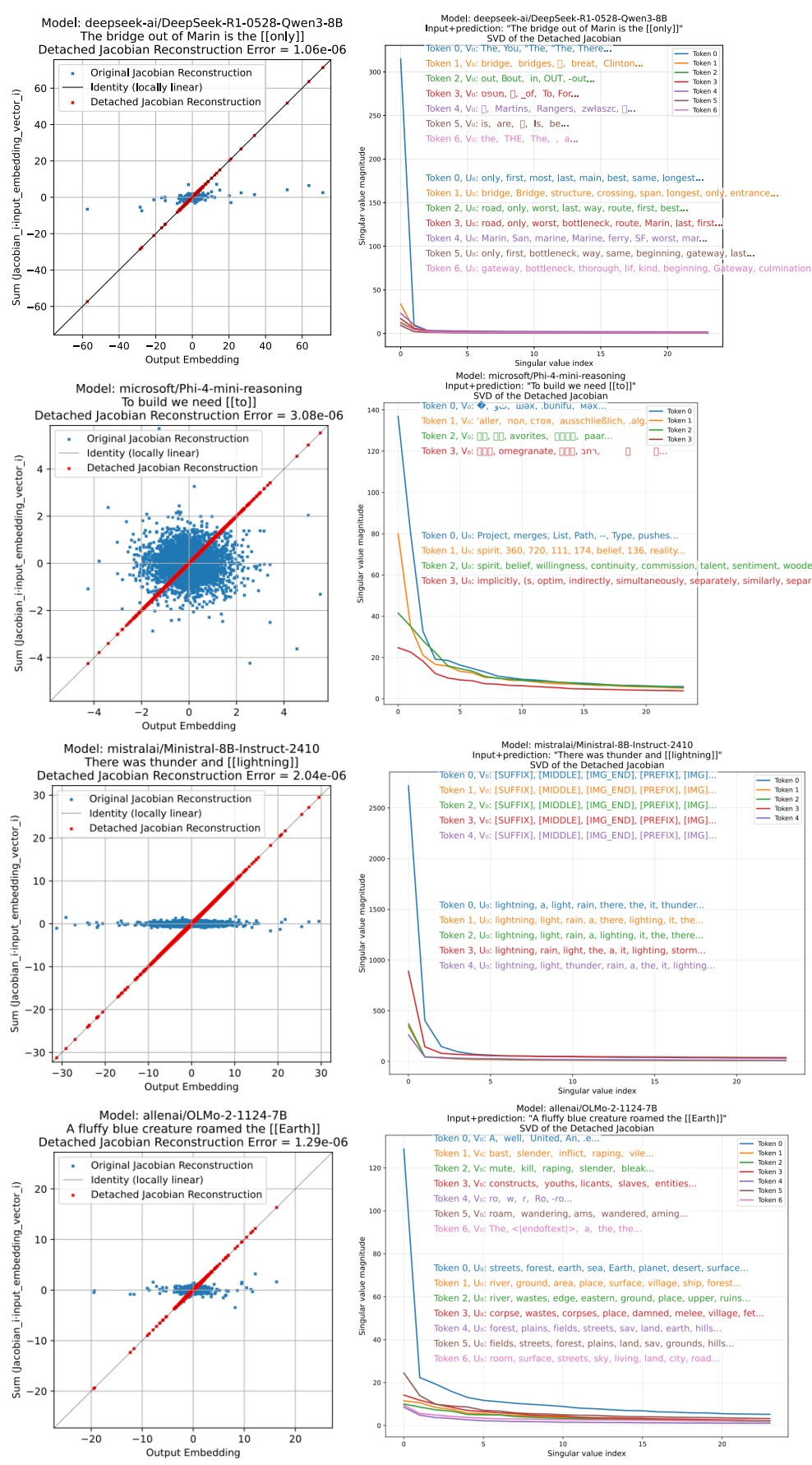

Figure A4: The detached Jacobian reconstruction error and SVD for Deepseek R1 0528 Qwen 3 8B, Phi 4 Mini 4B, Mistral Ministral 8B and OLMo2 7B.

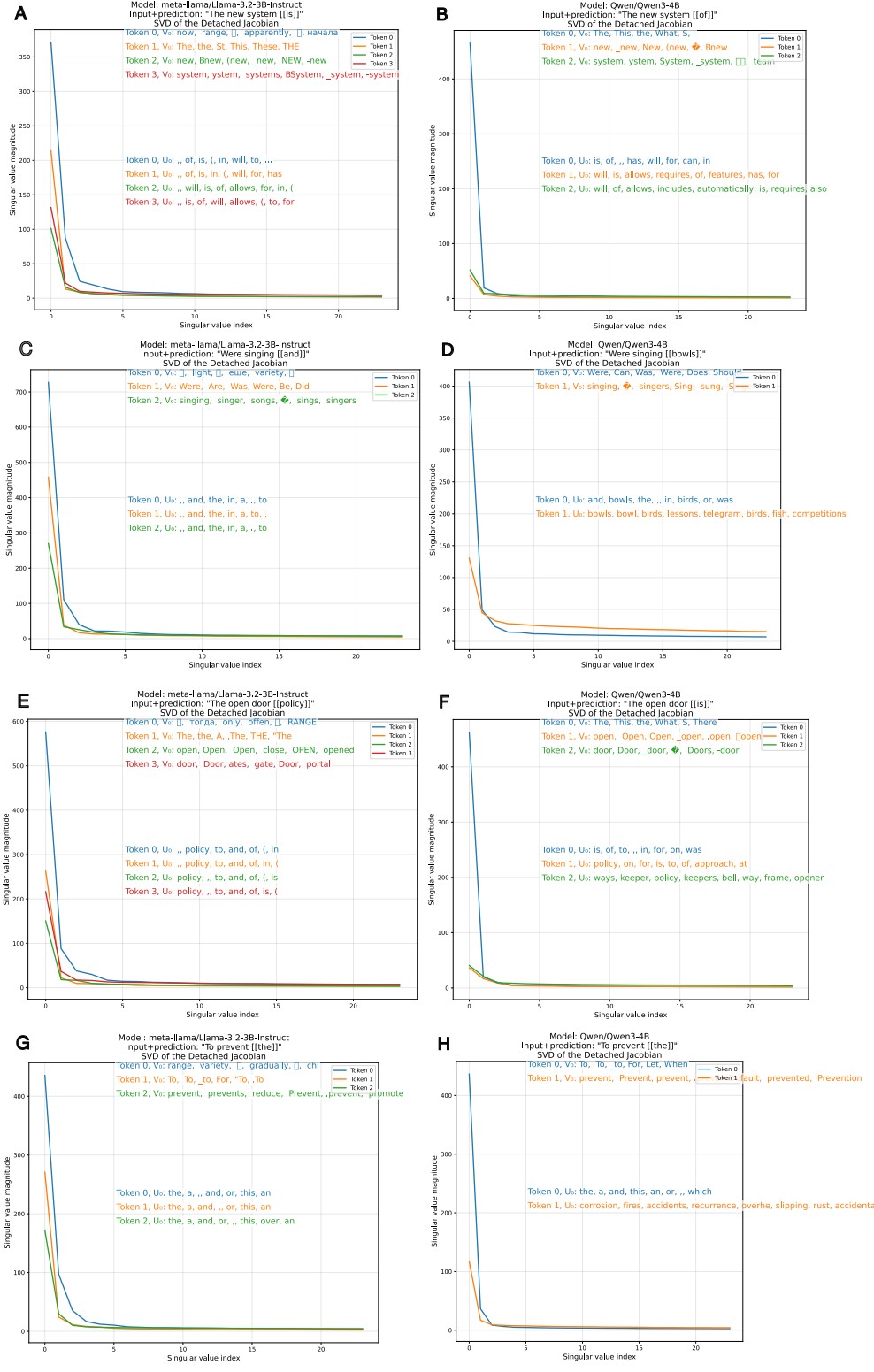

Figure A5: Comparison of detached Jacobians for the same phrase across models.

| | Input token 0 | Input token 1 | Input token 2 |
|---|---|---|---|
| Layer 13_0 | coli gnu ovny elin ovol ATEG | /Dk MetroFramework olumn reluct upertino DOT regor | akra izik esteem critical timer noch MUX apest |
| Layer 13_1 | weit LineStyle fonts Ymd ysize rt akra reverted | ta .unsplash fonts reverted Ymd ograd .gf | Lud QObject darwin adecimal )const angel PushButton usercontent |
| Layer 13_2 | chter i Burgess Lud abet Burke ernal *backslash*Bundle | opia agn ember missile trace osed | card plus imm cardinal Spare enz Eg lex |
| Layer 14_0 | en wil lo ... 764 fa | /Dk HeaderCode [OF ToPoint <typeof spd .liferay NCY | weit vc ciler inx critical akedirs |
| Layer 14_1 | weit ; dealloc akedirs LineStyle Bridge ysize | ta defs ones .unsplash arken ; | id rone orm iland [] .... |
| Layer 14_2 | chter Burgess <typeof Ridge Subsystem PressEvent | agn ember another arend | Kelly Jar Cunningham Jarvis )const Stadium ortal |
| Layer 15_0 | wil i Mage gnu iyah erval dedicated Mage | skb abus <typeof ToBounds xcf /Dk | bridge bridges Bridge Bridges Mood illin |
| Layer 15_1 | weit Bridge squeeze dealloc bridges woke bridge | illin k .foundation ophe ATEG bridges | gc RAD .bunifuFlatButton Dickinson PushButton NullException Comet |
| Layer 15_2 | chter <typeof RAD .NULL Subsystem Burgess ches | arend esser pair agn kel ered usi ending | Bravo Cunningham Imm Brew losures |
| Layer 16_0 | i en entre lei yre wil ewis iyah | sole /cms $MESS pdev xcf spd <typeof | bridge bridges Bridge k akra Bridges toll |
| Layer 16_1 | Bridge bridge squeeze akra SqlServer weit fabs | k 799 izi Bl akra . Exit | gc RAD Dickinson UpInside PushButton clerosis |
| Layer 16_2 | Invalidate RAD <typeof Subsystem /cms NUITKA saida | arend agn ag ered another /out dipl yll | vere losures Bravo Route employment closure exit occo |
| Layer 17_0 | i en yre iyah ewis erval only agnet | sole /cms gc Bridge bridge PushButton ToF | bridge Bridge bridges Bridges k choke Brig |
| Layer 17_1 | Bridge Marin bridge arLayout Brains bridges choke /connection | k yre .IDENTITY 799 Affected Bridges Local | gc PushButton ANNEL Inspiration bast occasion |
| Layer 17_2 | <typeof .scalablytyped Invalidate bast Burke gc | another isser hell to amp arend new gender | hosting closure Bravo ernal kond Hosting location Backbone |
| Layer 18_0 | bridge en i yre end 764 go San | bridge Bridge bridges sole crossing brid . bridge cause | bridge Bridge bridges Bridges brid Bridge bridge |
| Layer 18_1 | Bridge bridge bridges Marin Bridge Bridges brid | Bridges bridges OUNTRY .IDENTITY Queries Choices | PushButton gc Tos ANNEL Inspiration Route sole |
| Layer 18_2 | Invalidate .scalablytyped Burke Saunders stial Lair agues | to another elsewhere amp bec isser hell | San ernal hosting closure SF Bravo |
| Layer 19_0 | bridge SF exit San only connecting Oakland usp | bridge Bridge bridges crossing toll . bridge choke SF | bridge Bridge bridges Bridge Bridge bridge brid |
| Layer 19_1 | Marin Bridge bridge SF Golden Bridge bridge | Bridges Odds shima iliary ued IID Oakland .syntax | Highway Route route route ANNEL Hwy highway |
| Layer 19_2 | Fletcher Los LA Edgar Southern Burke LA Los | elsewhere to new Oakland another hell progress | SF San ucker Oakland Golden Stanford closure sf |
| Layer 20_0 | bridge toll exit Bridge bridges San SF Toll | bridge Bridge toll bridges Toll tol crossing . bridge | toll bridge Bridge tol latest bridges Toll Bridges |
| Layer 20_1 | Marin sf SF toll arLayout Oakland Berkeley | Odds Oakland contra n thing uckles ued istrator | Route route Highway Route odus route Hwy |
| Layer 20_2 | toFloat PLIED Southern LA Los Fletcher uluk bridge | to elsewhere new ella cht peninsula syn | SF Oakland Stanford anson San SF .vn sf |
| Layer 21_0 | bridge toll bridges exit San Bridge Toll SF | bridge toll Bridge bridges Toll tol Bridge crossing | bridge toll bridges Bridge Bridges tol Toll Bridge |
| Layer 21_1 | Marin SF sf SF Oakland Berkeley arLayout | contra n uckle Oakland thing | Route route .scalablytyped Highway odus Route annel |
| Layer 21_2 | uluk bridges bridge Bridge toFloat Southern PLIED Roose | Marin Oakland Fog ?type Berkeley SF uv | SF Marin Oakland SF San Stanford Berkeley sf |
| Layer 22_0 | bridge toll San bridges exit SF Bridge connecting | bridge bridges toll Bridge Bridges tol crossing Bridge | bridge bridges toll Bridge Bridges tol crossing Bridge |
| Layer 22_1 | Marin SF =īnas sf Berkeley | contra uckle thing 415 .ObjectId n | Highway Backbone Route highway route Route annel .Atomic |
| Layer 22_2 | fabs uluk bridges dealloc Brig GMT Bridge anlk | Marin ?type dea Berkeley arResult zc occo | SF Marin SF .sf sf SF Salesforce Berkeley |
| Layer 23_0 | toll San bridge exit Bay Golden Exit connecting | toll bridge Golden San Bridge tol bridges Toll | toll bridge bridges Bridge tol span Bridges Toll |
| Layer 23_1 | Marin inas =Ūkraj Berkeley | thing riz 415 orte contra uckle 299 imo | Pacific Route coast Highway route .Atomic annel |
| Layer 23_2 | fabs uluk Brig dealloc pun Roose Bud Cunningham | Marin ?type Berkeley dea zc ottenham inas aser | Marin SF Oakland .sf SF sf Berkeley Bay |
| Layer 24_0 | bridge San toll vi exit Bay SF Golden | toll bridge Golden San Bridge vi tol bridges | toll bridge span tol bridges Bridge vi |
| Layer 24_1 | Marin sf /goto marin aidu | riz ( contra rey ued uckle 299 | Pacific Highway Route Los .Atomic Los route highway |
| Layer 24_2 | uluk fabs dealloc anlk Brig Roose simd Bud | Marin dea ?type zc chez Berkeley app aidu | Marin SF .sf sf Oakland San SF Salesforce |
| Layer 25_0 | toll bridge vi subject connecting exit Bay only | toll bridge Golden vi tol Toll Bridge Golden | toll span bridge tol vi latest bridges Toll |
| Layer 25_1 | Marin aidu Skywalker arm sf | riz iny jev thing 415 contra | Pacific POSIT Via c Santa tracks rough Backbone |
| Layer 25_2 | uluk anlk fabs dealloc Brig GetEnumerator vka Roose | Marin chez dea ?type zc aidu ptr reib | Marin SF sf SF .sf San Richmond SF |
| Layer 26_0 | toll subject cause San Golden ge bridge only | Golden toll Bridge tol span cause Golden crossing | span toll tol Bridge spans Golden crossing vital |
| Layer 26_1 | Marin aidu .Generated mainwindow .scalablytyped | 415 arching iny n s contra riz | Pacific Via Santa rough rou annel Santa route |
| Layer 26_2 | uluk Brig anlk spans .Atomic fabs Bailey plied | Marin ptr dea ensis aidu chez lands inas | Marin sf SF .sf SF Richmond San 415 |
| Layer 27_0 | subject only cause SF ge exit connecting toll | span toll Golden cause tol San Toll only | span toll tol spans vital Span symbol latest |
| Layer 27_1 | Marin mainwindow /effects .Generated | arching n pone s yar | Pacific Santa .Atomic Santa annel route rou POSIT |
| Layer 27_2 | .Atomic uluk Bailey eview elig stdout ffset | Marin ensis zc dea chez lands agna | Richmond Marin SF sf .sf SF San SF |
| Layer 28_0 | Richmond only subject one last symbol toll I | Golden toll Richmond tol span San Toll only | span toll symbol tol latest Richmond only final |
| Layer 28_1 | P .Generated /effects | yar pone arching n s | Pacific .Atomic Santa Via Samuel twisting route |
| Layer 28_2 | uluk Bailey Ava ffset .Atomic jak Whip Santa | Marin odian dea ensis agna agnar zc olin | Richmond SF Marin SF sf .sf SF San |
| Layer 29_0 | only Richmond toll final last most one subject symbol | Golden most tol tol final Richmond Bay Golden | tol last final Golden symbol latest toll most |
| Layer 29_1 | Golden .Iter .Generated tslint | pone n arching beiter Ped yar jev | Via Santa Pacific Samuel Rim Corner winding coast |
| Layer 29_2 | Marin abra riad seite theless ordion sonian | Marin olin marin nov ensis aser agnar nov | Marin SF SF .sf Richmond sf San SF |

Table 3: The top three singular vectors of the detached Jacobian for the layer outputs from Llama 3.1 8B for the sequence "The bridge out of Marin is the" with the prediction [[Golden]]. Legend: "Bridge", "only", "highway", "exit", "most".

| | Input token 0 | Input token 1 | Input token 2 |
|---|---|---|---|
| Layer 13_0 | nt only the alors that ... and .. | the that only nt and most called either | probablement Guad alcanz Lans Yellow yaitu |
| Layer 13_1 | sh handful Shaq fame mselves pity mga | Sur GONE ). Ret Genau Ret | the and that the ... and if nt |
| Layer 13_2 | ively ional )]); | 0 5 1 2 4 mete 3 | Called March Bon Entity Clock Patricia Bin |
| Layer 14_0 | only that the nt ... one .. either | only the that one nt either only those | vecchio Bridge Guad probablement iconic menambah Night |
| Layer 14_1 | the 1 robot uvre | Sur GONE MaxLength Toto heus Seg Novo | the racist extremist that those terrorist scenic anarch |
| Layer 14_2 | Tyl | 0 1 5 2 4 3 verwenden 6 | and already after sure Z AL only development |
| Layer 15_0 | only called one probably either the very | only the one either that called probably transportation | iconic Bridge bridge IQR Nope puente bridges Bridges |
| Layer 15_1 | feminism 1 robot imperialism ems polticas | tzw Seg tzw North Sur Secondo aman atraves | one extremist racist the camping terrorist supposed military |
| Layer 15_2 | ) ). | 0 1 5 2 verwenden 3 4 6 | called Sept ( Grand |
| Layer 16_0 | only one very first the route | only the one first three two very | iconic Bridge bridge Centennial Greater Golden bridges Iconic |
| Layer 16_1 | tzw North bernama Nec tangent Bitter Seg getAvg | North North tzw tzw largest lagoon sogenannten shortcut | coastal scenic not military one likely very location |
| Layer 16_2 | Arc Approx | 0 5 1 2 3 Provides 6 4 | turbo Hydro Turbo Geo Mapa northward blasted north |
| Layer 17_0 | only first most one very two the | only one first the two most very | iconic legendary famed famous Greater namesake infamous Centennial |
| Layer 17_1 | hacerlo tangent tzw bernama north loophole | North north North tzw tzw sogenannte sogenannten behem | only very not scenic one likely military extremely |
| Layer 17_2 | ). Paths Tub .). ) Endpoints Heap | 0 1 5 2 3 Provides 4 verwenden | Turbo TOC Pipeline Aviation City Route Water |
| Layer 18_0 | only first most one very two the | only first one most two the that | bridge Bridge bridges iconic crossing Bridges |
| Layer 18_1 | loophole Nec Locator FBSDKAccessToken peanut sebaik | behem giant loophole lagoon swamp MaxLength Nec Locator | military only coastal area likely location beaches area |
| Layer 18_2 | ).. ). Paths | 0 Design 1 Style 3 Styling Provides | Water Pipeline Watercolor Pipelines Fountain Beacon Marathon Balloon |
| Layer 19_0 | only first one most two highway the | only one first two that most second | Bridge bridge bridges Bridges Crossing Bridge Golden crossing |
| Layer 19_1 | Ring Coc Road reverse Rd Beacon | Bridge Ring Coc Road notorious iconic behem Reverse | coastal military area location areas most beach beaches |
| Layer 19_2 | vreau rupani nggak inclusin advogado comprens emphas | Crossing Compre Laufe Steel Indem Cht Bridge | Aviation Outreach Pipelines Turbo Wastewater Pipeline Brewing Beacon |
| Layer 20_0 | one only most first second very | one only most first second best | Bridge bridge bridges Crossing Bridges crossing bridging iconic |
| Layer 20_1 | Mun Mun Har Tak Coc Trinity Beacon | Har Har Est Tak Mega Rainbow Coc | area one areas most location only coastal military |
| Layer 20_2 | Zap Typical Ric Tub +)$ Ridge | Bridge Crossing Bridges Crossing Bridge Guad | teapot minus Vectors Spiral spiral turbo spirals |
| Layer 21_0 | bridge only one most highway first best | bridge only one most first highway best | Bridge bridge bridges Bridge Bridges Crossing crossing bridging |
| Layer 21_1 | Bridge Bridge bridge bridges zungen Tol Tak | Bridge bridge Bridge bridge Bridges bridging Tol | most coastal area areas maritime one closest fastest |
| Layer 21_2 | vreau gobierno totalit comprens nggak lackluster ejrcito | Bridge Crossing Bridges Ponte Crossing Bridge | mansion Architectural Sculpture Basilica monumento Monument Museum edifice |
| Layer 22_0 | only bridge one most first highway new | only bridge one most first new highway | Bridge bridge bridges Bridges Bridge bridging Ponte |
| Layer 22_1 | Namara Puente chercher zungen McCullough klnb Wheeler | Bridge Puente puente bridges Tol Bridge | highway road route roads fastest most coastal coast |
| Layer 22_2 | ERISA vreau lackluster gacche pabbaj ceremonia prosa i | Bridge Bridges Bridge Geral Design | routes highways roads routes route ferries freeway expressway |
| Layer 23_0 | only bridge California most one highway freeway Bridge | only bridge California one most highway first new | bridge Bridge bridges Bridges Bridge bridging |
| Layer 23_1 | zungen Langer <unused58> qualiter loadNpmTasks menghilangkan | Bridge Bridges bridges <unused58> puente Bridge Langer | California freeway coastal Pacific trailhead fastest Californian route |
| Layer 23_2 | Sonoma Marin Napa Marin Esprito Medford California | Bridge Bridge Bridges bridge bridges | Sonoma Californians California Californian Valle Monterey Yosemite |
| Layer 24_0 | California only bridge freeway highway Highway Pacific Bridge | California only bridge freeway highway Highway Bridge one | bridge Bridge bridges Bridge Bridges bridge bridging puente |
| Layer 24_1 | <unused58> pored ! Bridges Langer | Bridges bridges Bridge Bridge puente <unused58> bridging | freeway highway route trailhead fastest roads highways pathway |
| Layer 24_2 | Sonoma Swiss Essex Marin Esprito Marin Medford | Bridges Bridge Bridge bridges bridges | routes Routes Route routes Route route route Routing |
| Layer 25_0 | bridge Bridge bridges California only bridge Highway highway | bridge Bridge bridges California only bridge ferry Highway | bridge Bridge bridges Bridge Bridges bridge bridging bridges |
| Layer 25_1 | Bridge Bridges Bridge bridges bridge bridging puente | bridges Bridge Bridges Bridge bridge bridging puente | highway route trailhead freeway roads road Highway trail |
| Layer 25_2 | Marin Marin Burmese SF SF Sonoma Genova | Bridges Bridge Bridge bridges bridges | Omaha Wichita Milwaukee Houston Memphis Chicago Nebraska Detroit |
| Layer 26_0 | bridge Bridge bridges California only San Highway most | bridge Bridge bridges California only San Highway most | bridge Bridge bridges Bridges bridge bridging bridges |
| Layer 26_1 | Bridges bridges Bridge puente bridge bridging | bridges Bridges Bridge puente bridge Bridge bridging | route highway trailhead freeway pathway Highway fastest road |
| Layer 26_2 | Marin Marin sf SF Burmese SF Sonoma | Bridge Bridge Bridges | Utah Angkor Boise Nebraska Alabama Omaha Mormon |
| Layer 27_0 | bridge bridges Bridge California only most Highway highway | bridge bridges Bridge California only most bridge Highway | bridge Bridge bridges Bridge Bridges bridge bridging puente |
| Layer 27_1 | Bridges puente bridges <unused25> Bridge | Bridges bridges puente bridging Bridge bridge Bridge | route highway trailhead pathway freeway most trail path |
| Layer 27_2 | Marin Marin Sonoma Burmese sf | Struct Structural Structural Struct | Utah Mormon Boise Angkor Alabama Cebu Birmingham Nebraska |
| Layer 28_0 | bridge Bridge only most California San Highway | bridge Bridge bridges only most California San Pacific | bridge Bridge bridges Bridge bridge Bridges bridging puente |
| Layer 28_1 | wachung oksatta athermy puente | puente Bridges bridges bridging <unused58> athermy | highway route pathway freeway Highway most path gateway |
| Layer 28_2 | Marin Marin Sonoma marin marin kafka | Struct Struct | Sonoma ruari yaml |
| Layer 29_0 | only most bridge one Golden California longest largest | only most bridge one Golden longest California largest | bridge Bridge bridges Bridge bridge puente bridging |
| Layer 29_1 | wachung azitt patx athermy orragie | puente bridging bridges Bridges Bridge | route highway trail path gateway pathway trailhead most |
| Layer 29_2 | Marin Marin Sonoma kafka Ukraj | <unused2146> struct Structure Struct Structural Structure | Snapshot nt Sonoma |
| Layer 30_0 | most one only bridge California new Golden | most only one California new bridge Golden longest | bridge Bridge bridges Bridge puente bridge toll |
| Layer 30_1 | wachung arakatuh Choibalsan athermy | bridging puente bridges getTransforms | route highway trail trailhead path pathway trails gateway |
| Layer 30_2 | Marin Marin marin Sonoma | struct Struct Structure | prescribe Snapshot nt |
| Layer 31_0 | most only one bridge new main Golden | only most one new main Golden bridge | bridge Bridge bridges Bridge bridge toll puente Toll |
| Layer 31_1 | bottlene Comunic azitt lytres qttr | puente bridges lytres | route trail highway path pathway trails trailhead Highway |
| Layer 31_2 | Marin Marin kuk | structures | Alabama Idaho Kansas Angkor Oklahoma Nebraska dunes fuselage |

Table 4: The top three singular vectors of the detached Jacobian for the layer outputs from Gemma 3 4B for the sequence "The bridge out of Marin is the" with the prediction [[Golden]]. Legend: "Bridge", "only", "highway", "exit", "most".

| | Input token 0 | Input token 1 | Input token 2 |
|---|---|---|---|
| Layer 20_0 | TRY NORMAL | massage | akedown eway slow congest nodeId |
| Layer 20_1 | AUSE nrw bbw metaphor .listFiles stret tgt | overlay extracts Liter | villa fashion getattr depress bias |
| Layer 20_2 | ade flutter Fil mon imm and ren | lyr bounding while | Entities campaign EventBus .FILL |
| Layer 21_0 | TRY REGARD dT | REGARD massage | eway slow exiting outbound fastest tight |
| Layer 21_1 | @end IGHL ocos UAGE crt | overlay substr tag adorn bestowed | Managed meds Choices TORT Madness machine Spare |
| Layer 21_2 | Terr tag iers imm Fil ues Mal | itol Tomorrow goodbye stash calar lyr syrup | HTTPS reinterpret UTF REFER JSON Netflix |
| Layer 22_0 | tweaking CONSTANTS | vacc getch Period | first hardest fastest exiting ramp |
| Layer 22_1 | metaphor unc DERP OBJC stret .wp ISP | substr MBOL bridge | hurry HIP opi Rockets TORT |
| Layer 22_2 | alk ole ool ros angan icon vn | antics ikerrocking | backstory weblog SVG JSON INCIDENT |
| Layer 23_0 | salopes CONSTANTS getch Uncomment massage TRY | metaphor bridge largest easiest only first centerpiece | first hardest unc fastest highway bottleneck |
| Layer 23_1 | metaphor unc Derne makeshift OBJC | bridge Bridge . bridge bridge | scenes WithError opi Timing presets Entering |
| Layer 23_2 | ros lovers flutter | antics jams | weblog COMPONENT annot metaphor |
| Layer 24_0 | first third last most largest fourth culmination | metaphor largest centerpiece easiest first hardest bridge gateway | hardest first easiest fastest most ones same |
| Layer 24_1 | metaphor makeshift .wp REAK | brid bridge bridges bridge Bridge | scenes LocalStorage WithError |
| Layer 24_2 | flutter rosLingu | | phenomena puzz annot metaphor |
| Layer 25_0 | largest most first longest latest fastest last third | bridge bridges Bridge gateway | hardest ones exit easiest first most fastest highway |
| Layer 25_1 | bridge bridges Bridge Bridges brid | bridges bridge Bridge bridge Bridge | ( exit exit exits eternity . exit |
| Layer 25_2 | bridges bridge Bridge bridge parliament | Exit exit jams | INCIDENT symbolism |
| Layer 26_0 | first most largest last longest latest gateway only | bridge bridges metaphor gateway connecting | highway first exit ones last hardest roads |
| Layer 26_1 | bridge bridges metaphor Bridges Bridge | bridges bridge structures brid bridge | .charset jams Margins |
| Layer 26_2 | parliament structures bridges Parliament bridge | Exit exit choke Exit panicked | symbolism metaphor |
| Layer 27_0 | first last largest bridge longest most oldest latest | bridge bridges Bridge Bridges | last first exit highway bottleneck next road choke |
| Layer 27_1 | bridge bridges Bridge Bridges | bridges bridge Bridge bridge brid Bridge | EXIT exit exits ( exit |
| Layer 27_2 | bridge bridge bridges Bridge structures | Exit exit Exit exit . exit | incident EXTRA incidents |
| Layer 28_0 | bridge longest largest first busiest last oldest most | bridge bridges Bridge Bridge | highway exit bottleneck highways Highway last road exits |
| Layer 28_1 | bridge bridges Bridge Bridge | highway highways coast freeway roads road route | exit exits EXIT exit Exit |
| Layer 28_2 | bridge bridge bridges Bridge brid | Exit exit Exit exit exit exit | Saddam Mosul Kuwait incident metaphor |
| Layer 29_0 | bridge only fourth last third longest fifth most | bridge bridges Bridge Bridges | only last first highway third highways exit fourth |
| Layer 29_1 | bridge bridges Bridge Bridges | coast highway road driveway coastline roads highways freeway | exits exit EXIT |
| Layer 29_2 | bridge bridges bridge structures brid structure | Exit exit Highway Exit | Saddam Mosul Elvis metaphor incident |
| Layer 30_0 | bridge most longest fourth third last only fifth | bridge bridges Bridge Bridge | highway only bridge last first Highway road highways |
| Layer 30_1 | bridge bridges Bridges Bridge | coast freeway highway coastline road roads highways | bridge Bridge bridges bridge brid |
| Layer 30_2 | bridge structure structures bridges bridge brid | sail seab sailing Bermuda ship | Memphis Kuwait Jordan Saddam Iowa |
| Layer 31_0 | bridge most only last longest first third largest | bridge bridges Bridge Bridge | only last highway first bridge exit Highway most |
| Layer 31_1 | coast airlines Interior airline interior Lua Speedway | coast coastline coastal Coast route beach Coastal | bridge Bridge bridges bridge underwater brid |
| Layer 31_2 | bridge bridges bridge brid Bridge structure | ship sail sailing dock seab | Jordan Memphis Kuwait Mississippi |
| Layer 32_0 | bridge most only first last longest third largest | bridge Bridge bridges Bridge bridge | only last first highway most main route exit |
| Layer 32_1 | interior airline steam airlines Trail breed vacuum | coast coastal coastline route Coast Route beach | bridge span underwater connecting deck public member |
| Layer 32_2 | bridge bridge bridges Bridge brid Bridge | ship sail dock sailing seab | Kuwait Jordan Memphis Edmonton Nile |
| Layer 33_0 | only first last most third main second subject | bridge Bridge bridges Bridge only | only last first key main same most exit |
| Layer 33_1 | planet interior cabin floors roots | coast coastline coastal Coast route beach Coastal | span public member library platform floating intervening deck |
| Layer 33_2 | bridge bridge structure bridges brid Bridge | ship orbit aircraft sail vessel | Kuwait Nile Edmonton Saskatchewan Tulsa |

Table 5: The top three singular vectors of the detached Jacobian for the layer outputs from Qwen 3 14B for the sequence "The bridge out of Marin is the" with the prediction [[only]]. Legend: "Bridge", "only", "highway", "exit", "most".

