# OpenReview forum: "Equivalent Linear Mappings of Large Language Models"
_TMLR — Accepted by TMLR_

### Review · Reviewer_P4MZ · 2025-07-17

**Summary Of Contributions:**

The paper present a method of representation analysis for Transformer LLMs based on linearization.

Specifically, the method apply to Transformer models without bias parameters and where all non-linear functions (activations, normalization and attention) are in the form of f(x, ...) = x * g(x, ...). The authors observe that in the forward pass the non-linear gating function g can be replaced by a "detached" Jacobian of f, obtained by autodiff with cloning the inputs of g and not propagating gradient backwards (corresponding to using the Pytorch "detach()" method on the cloned tensors). This is in general not the true Jacobian of f, but according to the paper it provides a better approximation at the operating point compared to the true Jacobian, which only provides a first-order Taylor approximation.

The authors apply this method to multiple mechanistic interpretability analyses:
- Analyse how the input and output embeddings linearly relate to each other at a specific input sequence, including a token relevance analysis
- Analyse the approximate rank (computed by SVD) of the mapping, finding that it is very low.
- They perform a "conceptual steering" experiment, where they mix the linearization of the model at one sequence with the computation of another sequence, observing semantic transfer between the two sequences.

**Audience:**

Yes

**Broader Impact Concerns:**

Broader Impact Statement is not present, and in my opinion not necessary for this paper.

**Claims And Evidence:**

Yes

**Requested Changes:**

- Quantitative analyses over in-distribution and out-of-distribution datasets.
- More concise and readable tables and graphs
- Better theoretical justification for the detached Jacobian method
- Remove unsupported claims about safety applications, or support these claims by experimental evidence

**Strengths And Weaknesses:**

Strengths:
- Interesting method
- The conceptual steering experiment is potentially useful as a mechanism for fine-grained control over the model behaviours.

Weaknesses:
- There is not enough discussion of why the detached Jacobian specifically provides a better linearization than the true Jacobian
- The graphs and especially the tables contain too much text which is way too small to read, even at high zoom. This isn't just a font issue, these graphs and tables should be restructured to be more concise.
- All analyses seem to have been done on a single sentence in the main text of the paper, with a few more sentences in the appendices. A more quantitative study on datasets of non-trivial size (e.g. 1000 examples), both in-distribution and out-of-distribution for the models, should have been performed in order to make the results more robust.
- The conclusion mention possible safety applications w.r.t. singular vectors for bias, misinformation or toxic content, but these are not identified in the paper, and it's unclear if they can be identified. This seems like an overclaim.

EDIT:

The authors have largely addressed my concerns in the last revision of the manuscript.

---

> ### Author Response · Authors · 2025-08-09
> **Comments to Reviewer P4MZ**
>
> ## Theoretical justification: the detached Jacobian and the true Jacobian
>
> We thank the reviewer for bringing up this point and understand that it could be discussed more in the manuscript.
>
> In the context of homogeneous functions of order 1, like a deep network with ReLU activations, the true Jacobian generates exact reconstructions of the output (as in Mohan et al. 2019). However, for LLMs with gated linear activations and nonlinear attention blocks, the LLM forward function is not homogeneous of order 1.Therefore the true Jacobian generates inaccurate reconstructions. However, with the appropriate gradient detachments at inference, the LLM forward function can be made locally homogeneous of order 1, such that its “detached” Jacobian generates an exact reconstruction.
>
> The detached Jacobian captures the same type of representation that the true Jacobian captures in ReLU networks (an exact but simultaneously interpretable representation), but for more complex and highly nonlinear LLM architectures. The detached Jacobian extends the exact linear decomposition and interpretation capabilities from simple ReLU networks to modern transformer architectures by selectively and exactly linearizing the components that break homogeneity. This explanation will be added to the manuscript.
>
> ## More diverse dataset
>
> We thank the reviewer for this criticism. We understand the value in broader analysis, but this manuscript was intended to be a proof-of-concept of the mathematical tools for this type of interpretation, not an analysis requiring statistical validation.
>
> The core contributions are demonstrating that LLM inference can be exactly decomposed into linear operations through strategic gradient detachment, showing how to extend homogeneous function analysis from ReLU networks to different types of LLMs (Qwen 3, Gemma 3, Llama 3, Phi 4, Mistral Ministral and OLMo 2), and providing new linear systems tools (SVD analysis, exact steering) for understanding LLMs.
>
> The mathematical properties we demonstrate are deterministic rather than statistical. Exact reconstruction is a necessary outcome given the homogeneity of the detached Jacobian, and the SVD interpretability is a direct consequence of the exact linear representation.
>
> We are open to this if necessary, but ideally we would carry out this experiment as future work at scale once the mathematical foundations are established. This paper provides the essential theoretical and engineering contributions needed for such applications.
>
> ## Graphs and tables should be restructured
>
> We thank the reviewer for this constructive point. The graphs and tables will be redesigned to be clearer.
>
> ## Safety applications
>
> We thank the reviewer for this point and agree with them. We will remove mentions of safety given the lack of evidence.

---

> > ### Comment · Reviewer_P4MZ · 2025-09-02
> > **Response**
> >
> > I think the authors have addressed the choice of the detached Jacobian vs the Jacobian.
> >
> > However, I'm struggling to find the main factual claim that we are supposed to evaluate here. The claim that the detached Jacobian achieves perfect reconstruction (error of 10^-14, which is the limit of float64 precision) is trivial from its definition. The main interesting claim is the rank analysis of the SVD of the detached Jacobian, but this is done only for one sentence. The authors added section 3.4 which is supposedly based on more example, but there is no rank analysis there, and it's still seems to be based on only a handful examples.
> >
> > I think this paper needs to clearly state what the main research question it addresses and what are the answers to these questions.

---

> ### Author Response · Authors · 2025-09-02
> **Response to Reviewer P4MZ**
>
> We thank the review for this useful comment.
>
> The main research questions are:
> Can the complex, globally nonlinear computation of LLMs be decomposed into simple equivalent linear systems?
> Do these linear system reveal interpretable and steerable semantic structure in next-token prediction?
>
> We agree that the exact reconstruction is in a sense trivial from the definition, but we believe that our definition of the detached Jacobian is non-trivial.
>
> In terms of the rank of the detached Jacobian of the 100 additional examples for Llama 3.2 and Qwen 3, we have these results but need to quantify the rank measurements, and will provide these shortly. Qualitatively, the spectra all have the same profile as the examples in the manuscript where the first 5-10 singular values are much larger than the rest.
>
> Edit: after an initial run of measurements, the stable rank is extremely low, between 1.05 and 1.5, with 95% of the power in only one or the first two singular values. The statistics will be added to the manuscript for 100 examples for Llama and Qwen.
>
> The main factual claims are listed below and are all in the manuscript but could be made more explicit. These changes will be made in the next revision today.
>
> The claims:
> - LLMs have an equivalent linear representation for each input, revealed by the correct set of detach operations. This is a non-trivial mapping of a nonlinear system to a linear system.
> - This method has the unique benefits of being exact, not requiring any training and relative ease of interpretability. Circuit analysis is not exact and leaves open the possibility that some other nonlinear component is affecting the operation, and SAEs must be trained for each network layer and correlated with semantic labels and are still ultimately nonlinear.
> - The detached Jacobian linear representations are extremely low-rank and interpretable, both in terms of rows and columns of the detached Jacobian, as well as the left and right singular vectors. There was not reason to expect this to be the case a priori.
> - The detached Jacobian works for semantic/conceptual steering, which is both a practical demonstration of its utility and an orthogonal validation that the detached Jacobian captures semantic information
> - This is a mathematical proof of concept for the detached Jacobian, and we plan to carry out a large-scale analysis of the detached Jacobian across datasets and models in future work.
>
> An additional point that is not in the manuscript:
> - This is an appearance of Sutton’s “bitter lesson” in LLM interpretability. With this conceptual insight where we can map nonlinear LLMs to linear systems, we can simply compute linear systems at scale and analyze them. This is a method that works broadly across models but requires computation at scale. The detached Jacobian moves the problem of interpretability for LLMs to the problem of interpreting many linear systems at scale (say for every token prediction in a dataset, which is difficult but possible) versus less exact, human-engineered methods for nonlinear systems.

---

> > ### Comment · Reviewer_P4MZ · 2025-09-04
> > **Response**
> >
> > This revision largely addresses my concerns.
> >
> > I've updated my recommendation.

---

### Review · Reviewer_4yX4 · 2025-07-30

**Summary Of Contributions:**

The submission:
- Shows that inference in an LLM can be approximated linearly at any given point using a detached Jacobian.
- Presents how to use the linear approximation to extract potentially interpretable features from layers of the LLM.
- Shows how steering can be performed using the extracted features.
To summarise, an alternative approach to interpreting LLMs other than autoencoders is proposed and implemented.

**Audience:**

Yes

**Claims And Evidence:**

Yes

**Requested Changes:**

## Content
- Please change *local linearity* to *linear approximation at fixed point* or some other appropriate term as the use of local linearity in this context is non-standard and confusing.
- Would be useful to add why this is an improvement or what advantages it may offer over existing interpretability methods.
- I don't see the relevance of figure 2, as far as I understand, is are the equations above the plot which could be typeset in the document, savings space.
- Please focus on one or two examples in Figure 3 and limit it to one or two subplots with a more informative keys which demonstrate the main finding (LLM inference can be linearly approximated for a given point).
- Figures 4 and 5 are hard to read with the lists of words overlain on the plot. Please separate them out. Maybe combine figures 4 and 5 into one (they are very similar) and this would give the space to present the lists of words in a more legible format. The shared figure could have 2 or at most 4 subplots.
- Axis label of figure 6B is not visible. Please fix.
- Tables 1,2, and 3 are too small to be legible it would be better if only a subset of the input tokens, as well as a subset of the layers were presented which would greatly improve legibility.
- Typo on page 11 line 2 (main text, not caption) - should be rest not rets.

**Strengths And Weaknesses:**

## Strengths

- The paper presents a new approach to interpretability of LLMs, that does not rely on training a separate layer (e.g. a sparse autoencoder) and then using an LLM to interpret the resulting features. Thus the features extracted correspond directly already existing embedding features, rather than automated labelling by an LLM. It proposes approaches to analyse and steer features.
- Good explanation of method and how to detach particular parts of the inference computation to obtain a linear approximation of the LLM inference function for a given input.
- Implementation provided that shows the approach works for a wide selection of open-weights LLMs.

## Weaknesses
- The paper uses the term local linearity, which commonly refers to a function being well approximated by a tangent line in the neighbourhood of a certain point. In the paper, however, the terms seems to mean a linear approximation of a non-linear function that is valid for a particular point. Thus while the main point paper, that LLM can be approximated linearly at each point holds, referring to this as a locally linear mapping seems misleading.
The authors themselves point out that: *This numerical Jacobian computation captures the complete forward operation of the model, including activation functions and attention, although it is only valid for that particular input sequence (more “pointwise” linear than
piecewise linear)*.
- The figures are cluttered and it is hard to understand them given the captions provided. Figure 3 attempts to show that the detached Jacobian is a good linear approximation of inference, but has 6 small subplots for different models, rather than 1 or 2 which would be a sufficient example.
- Overall it seems like the paper is really trying to fit within a 12 page limit, which is harming legibility and clarity.

---

> ### Author Response · Authors · 2025-08-08
> **Comments to Reviewer 4yX4**
>
> ## Change "local linearity" to "linear approximation at fixed point"
> We thank the reviewer for this insightful point regarding the term "local linearity." We agree that our terminology could be improved, as the method's validity is for a specific point, not a local neighborhood. We will use “point-wise” or “fixed point” and make a change to the title as well.
>
> The reviewer’s comment also prompted us to investigate the numerical precision of the detached Jacobian reconstruction more carefully. We conducted additional experiments for Llama 3.2 3B and Gemma 3 4B at float64 precision and found that the detached Jacobian achieves machine-precision exact reconstruction (torch.allclose=True), with relative errors at the level of machine epsilon or floating-point roundoff (~$10^{-16}$ for float64). This is a stronger result than our original 'near-exact' claim and demonstrates that we are computing an exact linear decomposition rather than an approximation. As described in the manuscript, this exactness is theoretically expected, since by detaching gradients from all nonlinear terms, we force the network to be homogeneous of order 1, which guarantees that the function can be exactly represented as $f(x) = J⁺(x) · x$ for the detached Jacobian $J⁺(x)$ computed at the operating point.
>
> We propose to use the more precise terms “pointwise linear equivalent” and "exactly equivalent linear system at a fixed point" throughout the revised manuscript. This terminology accurately captures both the point-specific nature of the Jacobian and the exact nature of the reconstruction.
>
> ## Focus on one or two examples in Figure 3
>
> We appreciate this point and apologize for the cluttered appearance of some of the figures and tables. We will prepare a new version of Figure 3 showing the exact reconstruction for  Llama 3.2 3B and Gemma 3 4B at float64 precision.
>
> ## Advantages the detached Jacobian may offer over existing interpretability methods
>
> We thank the reviewer for this suggestion and we will add a section on this in the Discussion. These are the points we have in mind in favor of the detached Jacobian.
>
>
> Mathematical exactness: captures the complete nonlinear computation as an equivalent linear system exactly in float64, whereas SAEs have reconstruction loss.
>
> No training required: computed directly from the existing model weights using automatic differentiation; SAEs must be trained for each layer of interest for each model, and linear probes must be trained on labeled data.
>
> Direct interpretability: singular vectors map directly to input/output token embeddings, providing immediate semantic interpretation without additional models; SAEs require another model for labeling semantic concepts.
>
> Scale and generality: works across model families (Llama, Gemma, Qwen) and at scales tested up to 70B parameters with identical methodology. SAEs are model-specific and circuit discovery is limited to small models.
>
> Steering without training: enables immediate extraction and injection of semantic concepts into unrelated sentences.
>
> ## Relevance of figure 2
>
> We agree with the reviewer and thank the reviewer for this point which will save space. We will remove figure 2 from the main text and typeset the equations.
>
> ## Figures 4 and 5
>
> Figures 4, 5 and 6 will be improved according to the reviewer’s suggestions.
>
> ## Tables 1, 2, and 3
>
> Tables 1, 2, and 3 will be improved according to the reviewer’s suggestions.
>
> ## Typo on page 11 line 2
>
> The typo will be corrected. We thank the reviewer for their constructive suggestions.

---

> > ### Comment · Reviewer_4yX4 · 2025-08-16
> >
> > Thank you for taking the suggestions into account. I look forward to seeing the final version of the paprer.

---

### Review · Reviewer_QBRf · 2025-08-03

**Summary Of Contributions:**

This paper defines the notion of the detached Jacobian (for a particular input sequence) as a tool for analyzing the behavior of LLMs. Given a particular input sequence, the detached Jacobian is constructed by “detaching” (i.e., telling the automatic differentiation library that certain variables be treated as constants for the purposes of differentiation) nodes of the computation graph such that the Jacobian of modified graph perfectly reconstructs the output of the original function on that input sequence. The high level idea is that given a non-linear operation in the graph, e.g., $g(x) = \sigma(x) \cdot x$, is can be replaced with a linear variant $f(x’) = \sigma(x) \cdot x’$ where $\sigma(x)$ is treated as a constant and thus the Jacobian at any $x’$ is $\sigma(x)$. When considering the original input, $x = x’$, $f(x)$ is a perfect reconstruction of the original output $g(x)$. The authors use this to (i) correlate high-magnitude rows in the detached Jacobian to particular tokens that follow logically from the original sentence, (ii) correlate singular vectors of the detached Jacobian with tokens, (iii) show the low (stable) rank of the detached Jacobian, and (iv) explore the detached Jacobian as a tool for output steering.

Overall, I think the detached Jacobian is a potentially promising tool for interpretability, but the paper needs to further strengthen the theoretical justification of this object and its empirical analysis.

**Audience:**

Yes

**Broader Impact Concerns:**

N/A.

**Claims And Evidence:**

No

**Requested Changes:**

1. **Further theoretical justification for the detached Jacobian in relation to the Taylor approximation**. The section above has some ideas on possible experiments/theoretical arguments that could be made. Addressing this would be critical in my opinion.
2. **A more comprehensive, quantitative experiment that considers other prefixes** (see section above for one possibility). I think this might not be critical in the presence of a very compelling theoretical justification for the detached Jacobian, but strong empirical insights could be used to argue for the usefulness of the detached Jacobian in lieu of a rigorous theoretical argument.
3. **Clarification on Section 3.6**. This would strengthen the paper, but it might not be critical (I would still appreciate clarification on the setup during the rebuttal).

**Strengths And Weaknesses:**

## Strength: Detached Jacobian as an analysis tool

The linearization of an LLM, and the analysis of such linearization, is an interesting idea for interpretability. There is a rich literature on linear maps which can be leveraged in such cases (e.g., the authors gesture to Lanczos-based approach from numerical linear algebra for more efficient SVDs). As far as I know, the definition of the detached Jacobian and its analysis is novel.

## Weakness 1: Theoretical justification of detached Jacobian

In my opinion, there are a couple of shortcomings with the definition and the detached Jacobian and its analysis:

**There is insufficient discussion and justification of this method in relation to a Taylor approximation**. One can also linearize a (non-linear) function at a particular input $x$ via Taylor’s theorem, which yields the approximation $f(x’) = f(x) + J(x) (x’ - x)$. This is also exact at $x’ = x$, and much studied than the detached Jacobian (e.g., bounded error in the neighborhood of x). I think it is fine to introduce a new notion, but there should be an elaboration on why, e.g., if you *require* a linear map $f(x’) = L(x) x’$, as opposed to the Taylor approximation’s affine map, this should be justified.

One experiment that might help justify the detached Jacobian is comparing quality of the approximation of the detached Jacobian vs. the true Taylor approximation in progressively larger neighborhoods around $x$. Showing they are very similar would suggest they are both valid ways of doing an approximation of an LLM at some input.

**The perfect reconstruction of the detached Jacobian might be trivial.** Figure 3 shows that the detached Jacobian “nearly exactly match the predicted embedding”. This is true, and the errors here ($10^{-3}$ for float16 and $10^{-6}$ for float32) suggest they are equivalent up to machine precision. However, as far as I understand, the detached Jacobian is *defined* to perfectly reconstruct the output for a particular input, so this result is unsurprising. Moreover, because of what is mentioned above, we know a Taylor approximation would also satisfy this property, though the function here is affine. So the fact that the (true) Jacobian is presented as having large error feels slightly misleading, unless you can justify the need for a linear map (i.e., compared the true vs. detached Jacobians without any bias).

As an extreme example, we know that the detached Jacobian at a point $x$, $D$, is a linear map that satisfies $D x= f(x)$, but there infinite linear maps satisfying such a property, so Figure 3 would remain exactly the same for all those alternative definitions of a detached Jacobian, though obviously the subsequent analysis would change. Why is it that the detached Jacobian *specifically* is the right construction? Intuitively it feels “closer” to the way the LLM operates, but hopefully there is a more grounded way to justify this.

## Weakness 2: More comprehensive and clearer empirical evaluation

The results of the empirical evaluation are interesting, but they feel limited in scope. As far as I can tell, sections 3.2, 3.3, 3.4 and 3.5 consider a single sentence “The bridge out of Marin is the”. While delving deep into one example is good, the paper would benefit from a more comprehensive quantitative evaluation. For example, for comparisons across model families, you could take 100-1000 sentences and see how similar the predictions from the singular vectors/large magnitude rows/etc. are across model families.

I found Section 3.6 but a bit hard to follow. I think there should be more explanation of the method and empirical setup.

---

> ### Author Response · Authors · 2025-08-06
> **Comments to Reviewer QBRf**
>
> # Theoretical justification of detached Jacobian
>
> ## Taylor approximation
>
> We appreciate the reviewer's thoughtful question about the relationship between our detached Jacobian approach and Taylor approximation. This comparison highlights a fundamental difference in interpretability objectives that we would like to clarify.
> Our approach builds on established theory for homogeneous functions of degree 1, following Mohan et al. (2019) and their application of Euler's theorem. For such functions, $f(x) = J(x)·x$ exactly, where $J(x)$ is the Jacobian. This provides a clean decomposition where the entire function output is explained purely through linear operations on the inputs, with no residual constant or bias term.
>
> The key advantage of the homogeneous function approach over Taylor approximation lies in interpretability. Consider the two approaches:
>
> Taylor approximation:                 $f(x₀ + h) ≈ f(x₀) + J(x₀) · h$
>
> Homogeneous decomposition:    $f(x₀) = J⁺(x₀) · x₀$
>
> In Taylor approximation, at the operating point ($h = 0$), the function value $f(x₀)$ comes entirely from the constant term, while the Jacobian contributes nothing (since $J(x₀) · 0 = 0$). This means the Taylor approach cannot explain how the input x₀ produces the output $f(x₀)$ through interpretable linear operations.
>
> The detached Jacobian $J⁺(x₀)$ directly shows how each input embedding contributes linearly to reconstruct the output embedding. This enables interpretability analyses like singular value decomposition of the linear operators, decoding of feature directions to tokens (as shown in our Tables 1-3), and layer-by-layer analysis of concept emergence.
>
> The discrete nature of token embeddings makes continuous neighborhood analysis less relevant for LLM interpretability. Since token embeddings occupy specific points in the continuous space, we are interested in understanding the network's behavior at those discrete locations rather than in continuous neighborhoods. Our method provides nearly exact reconstruction at these points of interest, which is most relevant for LLM analysis. For the Taylor approximation, this is the $h = 0$ case, where $J(x₀) · 0 = 0$ and the approximation comes only from the constant term, so the true Jacobian does not hold much utility for interpretation.
>
> ## The perfect reconstruction of the detached Jacobian might be trivial
>
> We thank the reviewer for this comment. Figure 3 is a numerical demonstration of the local homogeneous function equality, and evidence that gradient detachment enables near-exact reconstruction with a linear system. It was not clear in principle that strategically detaching gradients from normalization layers, gated linear activations (SwiGLU, GELU), and the softmax attention blocks would yield linear operators that exactly reconstruct outputs. This required numerical confirmation of the accuracy of the detached Jacobian reconstruction which resulted in Fig. 3.
>
> ## Why is it that the detached Jacobian specifically is the right construction?
>
> We thank the reviewer for this crucial question about the faithfulness of the detached Jacobian to the LLM forward function. We agree that infinitely many linear maps could satisfy $D · x$*$  = f(x$*$)$, but the detached Jacobian is uniquely determined by the network's computational structure.
>
> The computation of the detached Jacobian is not arbitrary. It is a linear map that preserves the computational path of the original network. The detached Jacobian $J⁺(x*)$ is computed via autograd, making it the unique linear operator that follows the network's actual computational graph with nonlinear terms frozen at their evaluated values. Unlike arbitrary linear maps, this construction also works for all of the network's intermediate operations, including attention blocks and MLP blocks, which all have linear equivalents that can be computed and reproduce the intermediate representation. The detached Jacobian simultaneously satisfies reconstruction at every intermediate layer: $y₁$*$ = J₁⁺(x$*$) · x$*$, y₂$*$ = J₂⁺(x$*$)·x$*$, ..., y_n$*$ = J_n⁺(x$*$)·x$*. This set of constraints dramatically reduces the solution space.
>
> Additionally, for single-token inputs, the reconstruction becomes a true linear system (not multilinear), which has a unique solution given the intermediate layer constraints. The detached Jacobians of this transform can be computed for each individual layer, and compose the whole-network Jacobian correctly as $J⁺(x$*$) = J_{0:1}⁺(x$*$)  · J_{1:2}⁺(x$*$) · … · J_{N-1:N}⁺(x$*).
>
> We have also demonstrated that the detached Jacobian is a general method that finds this linear system for any input sequence over a range of models.
>
> # Requested Changes
>
> We sincerely thank the reviewer for careful examination of the manuscript and will consider what we can do in terms of these suggestions. We will provide a more detailed plan of our changes shortly (inc. S3.6), and would greatly appreciate the reviewer’s feedback on the points detailed above.

---

> ### Author Response · Authors · 2025-08-06
> **Comments to Reviewer QBRf (continued)**
>
> # Clarification on Section 3.6.
>
> Section 3.6 demonstrates how the detached Jacobian from a middle layer can serve as a steering operator to influence model outputs toward specific concepts. The general idea is that by actively engineering outputs with the detached Jacobian, we can further demonstrate its utility, and validate that it truly captures semantic information. We hope that the success of these experiments serves as further support for the success of the approach, perhaps in lieu of the singular vector analysis of a more diverse set of sentences. The key insight is that once we compute the detached Jacobian $J⁺_L(x_steer$*) for a "steering concept" (e.g., "Golden Gate Bridge"), this linear operator captures how input embeddings should be transformed to produce that concept at layer L.
>
> For a new input sequence $x_{new}$*, we can steer the model by:
>
> Computing the normal layer L activation: $f_{L}(x_{new}$*)
>
> Applying the steering operator: $J_{L}⁺(x_{steer}$* $) · x_{new}$*
>
> Taking the weighted combination: $λ · f_{L}(x_{new}$* $) + (1-λ) · J_{L}⁺(x_{steer}$* $) · x_{new}$*
>
> This steered representation is then fed through the remaining layers to generate the final output.
> These details can be added to the manuscript.
>
> # Weakness 2: More comprehensive and clearer empirical evaluation
>
> We acknowledge with the reviewer’s concern that our empirical evaluation is too narrow. Focusing on a single input sequence is useful for deep analysis across models (with singular vector analysis, layer-by-layer decomposition and steering, demonstrating a wide range of applications of the method). The hope was that a single sentence would reveal interesting cross-model comparisons. We will attempt to work on a larger set of sentences with results to include in the paper.

---

> > ### Comment · Reviewer_QBRf · 2025-08-29
> > **Response**
> >
> > I thank the authors for their response.
> >
> > I really appreciate the explanation regarding the detached Jacobian. Upon re-reading the paper and going through Mahon et al. (2019), that concern is addressed.
> >
> > While new results are shared in the supplementary material, it seems these haven't been integrated into the paper, so this remains unaddressed. I believe that a systematic, quantitative evaluation would really help determine the usefulness of the method, so I would encourage the authors to reconsider their stance and improve this in the final version of the paper. I concede that this point may be subjective, however.
> >
> > I do still think that the reconstruction results are redundant: since you define the detached Jacobian such that the reconstruction is perfect, I don't see how these errors are anything but rounding errors (and the Jacobian is obviously going to underperform). While not a problem per se, I again think the space could be better used for more quantitative evaluation.

---

> ### Author Response · Authors · 2025-08-30
> **Comment regarding new section added with results comparing across models**
>
> We thank the reviewer for this additional comment to the revision.
>
> In response, based on experiments with 100 examples for Llama 3.2 3B and Qwen 3 4B, we have added section 3.4 "Analysis of the SVD across models" and Fig. A4 in the appendix. The raw results from the examples are also included in the supplementary data.
>
> At this time we have not removed the figure with reconstruction error, but would be amenable to that if needed (currently there is still space as is).
>
> Note: As of Sep 1, this section has been updated and expanded on with a number of specific examples for each observed principle in the appendix.

---

### Decision · Action_Editor_ugxu · 2025-09-04

**Recommendation:** Accept with minor revision

**Audience:**

Yes

**Audience Explanation:**

This is broadly interesting to people who study the behaviour and theory of neural networks.

**Claims And Evidence:**

Yes

**Claims Explanation:**

While the reviewers had some doubts about the paper's validity in an early round, after the author revised the paper and addressed issues with the experiments and explanations regarding the detached Jacobian, the reviewers unanimously decided that the paper has a place in TMLR. I support their decision and believe the paper satisfies the soundness acceptance criterion of TMLR.

There is still an issue lingering with the scope of the experiments, stated by the reviewer: "... The claim that the detached Jacobian achieves perfect reconstruction (error of 10^-14, which is the limit of float64 precision) is trivial from its definition. The main interesting claim is the rank analysis of the SVD of the detached Jacobian, but this is done only for one sentence. The authors added section 3.4 which is supposedly based on more example, but there is no rank analysis there, and it's still seems to be based on only a handful examples." I hope the author can address as a minor revision.

**Resubmission Of Major Revision:**

The authors may consider submitting a major revision at a later time.